# Direct assimilation of ground-based microwave radiometer observations with machine learning bias correction based on developments of RTTOV-gb v1.0 and WRFDA v4.5

Qing Zheng[1,3], Wei Sun[2,3], Zhiquan Liu[4], Jiajia Mao[5], Jieying He[6], Jian Li[2,3], Xingwen Jiang[1,3]

[1]Heavy Rain and Drought-Flood Disasters in Plateau and Basin Key Laboratory of Sichuan Province, Institute of Tibetan Plateau Meteorology, China Meteorological Administration, Chengdu, 610213, China
[2]State Key Laboratory of Severe Weather Meteorological Science and Technology (LaSW), Chinese Academy of Meteorological Sciences (CAMS), Beijing, 100081, China
[3]Institute of Tibetan Plateau Meteorology, Chinese Academy of Meteorological Sciences, Beijing, 100081, China
[4]National Center for Atmospheric Research, Boulder, 80307-3000, USA
[5]Meteorological Observation Center, China Meteorological Administration, Beijing, 100081, China
[6]National Space Science Center, Chinese Academy of Sciences, Beijing, 100190, China

*Correspondence to*: Wei Sun (sunwei@cma.gov.cn)

**Abstract.** The application of ground-based microwave radiometers (GMWRs), which provide high-quality and continuous vertical atmospheric observations, has traditionally focused on the indirect assimilation of retrieved profiles. This study advanced this application by developing a direct assimilation capability for GMWR radiance observations within the Weather Research and Forecasting Data Assimilation (WRFDA) system, along with a bias correction scheme based on the random forest technique. The proposed bias correction scheme effectively reduced the observation-minus-background (O−B) biases and standard deviations by 0.83 K (97.1 %) and 1.63 K (64.6 %), respectively. A series of ten-day experiments demonstrated that assimilating GMWR radiances improves both the initial conditions and the forecasts, with additional benefits from higher assimilation frequencies. In the initial conditions, hourly assimilation significantly enhanced low-level temperature and humidity fields, reducing the root-mean-square error (RMSE) for temperature by 6.32 % below 1 km and for water vapor mixing ratio by 1.98 % below 5 km. These improvements extended to forecasts, where 2 m temperature and humidity showed sustained benefits for over 12 h, and precipitation forecasts exhibited improvements to a certain extent. The time-averaged Fractions Skill Score (FSS) for 3 h accumulated precipitation within the 24 h forecasts increased by 0.02–0.04 (3.9–10.2 %) for thresholds of 3–6 mm.

## 1 Introduction

Data assimilation (DA), a core component of numerical weather prediction (NWP), plays an important role in improving forecast accuracy by integrating observational data to refine initial conditions (Bauer et al., 2015; Gustafsson et al., 2018). Among various types of observations, microwave radiance data are crucial for DA due to their ability to penetrate the

atmosphere and their sensitivity to temperature, humidity, clouds, and precipitation. Correspondingly, satellite-borne microwave radiance observations have been extensively studied and are considered among the most influential contributors to data assimilation systems (Geer et al., 2017; Kim et al., 2020; Candy and Migliorini, 2021).

Unlike satellite-borne microwave radiometers, ground-based microwave radiometers (GMWRs) offer unique advantages for DA, including high temporal resolution (minute-level) and greater sensitivity to the atmospheric boundary layer (ABL). Over the past two decades, the assimilation of GMWRs has been increasingly studied, leading to improvements in the accuracy of NWP (Vandenberghe and Ware, 2002; Otkin, 2010; Hartung et al., 2011; Otkin et al., 2011; Caumont et al., 2016; He et al., 2020; Qi et al., 2021, 2022; Lin et al., 2023). The assimilation of retrieved temperature and humidity profiles from GMWRs

has shown improvements in forecasting fog, storms, and precipitation. However, the reliance on indirect assimilation methods introduces uncertainties and complicates error quantification, which limits their overall effectiveness in enhancing forecast accuracy (Caumont et al., 2016; Martinet et al., 2017; Lin et al., 2023).

Direct assimilation of GMWR radiances, which bypasses the retrieval process, offers significant advantages by avoiding retrieval-related errors and improving the effective use of observations. This approach requires accurate observation operators

and robust bias correction to address differences between radiance observations and model states. The direct assimilation of satellite-borne radiance observations is relatively mature (Geer et al., 2008; Bauer et al., 2010; Geer et al., 2010; Eyre et al., 2020; Sun and Xu, 2021; Eyre et al., 2022) and utilizes fast radiative transfer models (RTMs) as observation operators, such as the Radiative Transfer for Television and Infrared Observation Satellite (RTTOV) (Saunders et al., 2018). However, the unique characteristics of upward-looking GMWR observations, such as sensitivity to near-surface conditions, require

specialized RTMs and adaptation of existing techniques. Recent studies have developed fast RTMs suitable for GMWRs, which provide a foundation for constructing observation operators for assimilation of GMWR radiances (De Angelis et al., 2016; Cimini et al., 2019; Shi et al., 2025). The RTTOV-gb, a ground-based version of the RTTOV model, was used to simulate brightness temperature from GMWRs, demonstrating high accuracy (De Angelis et al., 2016, 2017; Cimini et al., 2019). Recent studies have demonstrated the potential of direct GMWR radiance assimilation using RTTOV-gb to improve temperature,

humidity, and precipitation forecasts (Cao et al., 2023; Vural et al., 2024).

Despite these advancements, previous studies have typically relied on limited GMWR networks or focused on specific case studies. Additionally, research conducted in regions with relatively simple terrain may not fully address the complexities of areas like the Tibetan Plateau, where complex topography often leads to significant model biases (Yang et al., 2020; Wei et al., 2021). These biases make accurate bias correction essential for improving the effectiveness of direct assimilation, while

traditional bias correction approaches developed for satellite-borne microwave radiance observations are not directly applicable to GMWRs.

To address these issues, this study integrates RTTOV-gb into the Weather Research and Forecasting Data Assimilation (WRFDA) system (Barker et al., 2012) to develop a direct assimilation module for GMWR radiances. A nonlinear bias correction scheme based on machine learning is also constructed using three months of observational data. The impact of direct

GMWR assimilation is then investigated through a series of ten-day experiments conducted in Southwest China, a region

shaped by the influence of the Tibetan Plateau and characterized by complex terrain. The remainder of this paper is organized as follows. Section 2 describes the data, the implementation of RTTOV-gb in WRFDA, and the model configuration. Section 3 evaluates the performance of the bias correction scheme. Section 4 presents the impacts of GMWR assimilation on the initial and forecast fields. The conclusions and discussion are presented in Sect. 5.

**2 Methodology**

**2.1 Data**

Two types of GMWR sensors were assimilated in this study, as shown in Fig. 1: the MP3000A and the Humidity And Temperature PROfiler (HATPRO). Atmospheric radiance is measured as brightness temperatures in 14 channels for HATPRO and 22 channels for MP3000A (Table 1). For HATPRO, channels 1–7 are in the K-band, while channels 8–14 are in the V-
band. For MP3000A, channels 1–8 are in the K-band, and channels 9–22 are in the V-band. The K-band channels correspond to humidity-sensitive water vapor absorption lines, whereas the V-band channels correspond to temperature-sensitive oxygen absorption lines.

The Fengyun-4B (FY-4B) Advanced Geostationary Radiation Imager (AGRI) cloud mask (CLM) is used to identify GMWR-observed brightness temperatures under clear-sky conditions. The AGRI-based CLM product has a temporal resolution of 15
minutes and a horizontal resolution of 4 km, categorizing conditions as confidently cloudy, probably cloudy, probably clear, or confidently clear, with corresponding values of 0, 1, 2, and 3, respectively (Min et al., 2017). Due to its high quality, this cloud mask product is widely applied in satellite data assimilation (Yin et al., 2020, 2021; Xu et al., 2023; Shen et al., 2024).

The National Centers for Environmental Prediction (NCEP) Final Operational Global Analysis data (FNL) (0.25° × 0.25°, 6-hourly) were used to establish the initial and boundary conditions for regional NWP. Conventional observations from the
Global Telecommunications System (GTS) were assimilated and evaluated, including land surface, marine surface, radiosonde, and aircraft reports. The hourly precipitation analysis product from the China Meteorological Administration Multisource Precipitation Analysis System (Shen et al., 2014) was used for evaluation. This dataset has been widely used in precipitation studies (Xia et al., 2019; Su et al., 2020; Sun and Xu, 2021; Wang et al., 2021; Li et al., 2023; Zheng et al., 2024).

Table 1. Central frequency for GMWRs

| Sensor | Frequencies for K-band (GHz) | Frequencies for V-band (GHz) |
|---|---|---|
| HATPRO | 22.240;23.040;23.840;25.440; 26.240;27.840;31.400 | 51.260;52.280;53.860;54.940; 56.660;57.300;58.000 |
| MP3000A | 22.234;22.500;23.034;23.834; 25.000;26.234;28.000;30.000 | 51.248;51.760;52.280;52.804; 53.336;53.848;54.400;54.940; 55.500;56.020;56.660;57.288; 57.964;58.800 |


## 2.2 Assimilation system and observation operator

The WRFDA system, developed by the National Center for Atmospheric Research (NCAR), is designed for data assimilation and includes three-dimensional variational (3DVAR), four-dimensional variational (4DVAR), and hybrid data assimilation algorithms. In this study, version 4.5 of the WRFDA system with 3DVAR is used for the direct assimilation of GMWR radiances. The 3DVAR algorithm produces the analysis by minimizing a scalar objective cost function:

$$J(x) = \frac{1}{2}(x - x_b)^T \mathbf{B}^{-1}(x - x_b) + \frac{1}{2}(y - \mathbf{H}(x))^T \mathbf{R}^{-1}(y - \mathbf{H}(x)), \tag{1}$$

where $x$ and $x_b$ represent the analysis and background fields of the model variables, $\mathbf{y}$ is the vector of the observations, and $\mathbf{B}$ and $\mathbf{R}$ represent the background and observation error covariance matrices, respectively. The covariance matrices determine the weights assigned to the background and observations in the analysis, dictate how localized observation information is distributed vertically and horizontally in the model space, and maintain the balance among the model's control variables. $\mathbf{H}$ is the nonlinear observation operator that transforms model variables to the observed quantities. The observation operator works slightly differently for different types of observations. For conventional observations (e.g., temperature), its primary role is to perform spatiotemporal interpolation of model grid values to the observation space. For unconventional observations (e.g., reflectivity and radiance), where the model state cannot be directly compared with the observations, the observation operator must also convert model variables into observed variables.

The static background error covariance for the variational experiments is estimated using the National Meteorological Center (NMC) method (Parrish and Derber, 1992), which uses the difference between WRF forecasts at lead times of 24 h and 12 h (T + 24 h minus T + 12 h) valid at the same time over a specified period. Control variables option 5 (CV5) is adopted for the background error covariance used in 3DVAR. CV5 is domain-dependent and therefore must be generated based on forecast or ensemble data over the same domain. It utilizes streamfunction, unbalanced velocity potential, unbalanced temperature, unbalanced surface pressure, and pseudo relative humidity. In this study, the background error covariance matrix was generated using the Generalized Background Error Covariance Matrix Model (GEN_BE v2.0) (Descombes et al., 2015) based on one month of WRF forecasts. Observation-error correlations are typically assumed to be zero in WRFDA, resulting in a diagonal observation-error covariance matrix. Observation errors were specified based on the standard deviation of O−B.

RTMs serve as observation operators for assimilating radiance data by mapping model variables (e.g., temperature and water vapor) into radiance space. RTTOV, a fast RTM, is widely used for assimilating satellite radiance data. However, GMWR radiances are upward-looking microwave observations, differing from the downward-looking measurements of satellite-borne microwave radiometers. This difference in direction makes RTTOV difficult to apply to GMWR radiance assimilation. Fortunately, RTTOV-gb can simulate brightness temperatures from GMWRs and serves as the observation operator in this study. The weighting function (WF) quantifies the contribution of emissions from each atmospheric layer, and the maximum WF height indicates which atmospheric layer contributes most to the measured radiance (Carrier et al., 2008). According to Cui et al. (2020), WFs are calculated as the derivative of transmittance with respect to the natural logarithm of pressure. The

vertical distribution of WFs for HATPRO and MP3000A, calculated using RTTOV-gb, is shown in Fig. 2a–b. The WFs reach their maximum at 1000 hPa and decrease monotonically with height. These results confirm that the lower atmosphere contributes most to the observed radiance across all channels, consistent with the findings of Shu et al. (2012).

It should be noted that RTTOV-gb is not included in the publicly available version of WRFDA. To address this limitation, a GMWR direct assimilation module was developed within WRFDA. A single-observation assimilation experiment was conducted to test the GMWR direct assimilation module. In this experiment, a pseudo radiance observation from the HATPRO sensor was assimilated at channel 6 (K-band), with an assigned observation error of 1 K and an innovation of 2 K. Results confirm that the GMWR direct assimilation module performs correctly. The temperature and water vapor increments are horizontally isotropic and show a maximum at lower atmospheric levels vertically (Fig. 2c–f). It should also be noted that this experiment was conducted to verify the correct performance of the GMWR direct assimilation module and to provide valuable insights into the characteristics of GMWR assimilation. However, it is not representative of the subsequent multi-observation, multi-channel assimilation experiments.

## 2.3 Model configuration and experimental design

In this study, version 4.5 of the Weather Research and Forecasting (WRF) model (Skamarock et al., 2021) is used to simulate atmospheric evolution. The simulation employs a single domain (Fig. 1) with a horizontal resolution of 3 km, comprising $1,261 \times 811$ grid points and 51 vertical levels, with the top boundary at 10 hPa. The model physics configuration includes the Morrison two-moment microphysics scheme (Morrison et al., 2009), the Yonsei University PBL scheme (Hong et al., 2006), the Rapid Radiative Transfer Model for General Circulation Models (RRTMG) shortwave and longwave radiation schemes (Iacono et al., 2008), and the unified Noah land-surface model (Chen and Dudhia, 2001). Cumulus parameterization was excluded due to the convection-permitting horizontal resolution of 3 km (Li et al., 2023; Moker et al., 2018).

Similar to previous studies (Jiang et al., 2017; Nie and Sun, 2023), the target region of Southwest China in this study is defined as the area within the rectangular domain 22–35° N, 93–110° E (Fig. 1). This region encompasses the Hengduan Mountains, the Yunnan–Guizhou Plateau, and the Sichuan Basin. Based on the model configuration described above, four parallel experiments were conducted to investigate the impact of GMWR assimilation (Table 2). Each experiment started at 12:00 UTC daily, followed by a 12 h cycling data assimilation period and a subsequent 24 h forecast. The primary differences among these experiments lie in the assimilated data and assimilation intervals. The CNTL experiment assimilated GTS data with a 6 h interval, while the GMWR_6H experiment added GMWR assimilation to the CNTL setup, enabling an evaluation of GMWR assimilation's impact. The other two experiments, GMWR_3H and GMWR_1H, assimilated both GTS and GMWR data with 3 h and 1 h intervals, respectively, to assess the effects of observation frequency in GMWR assimilation.

The assimilation experiments were conducted under clear-sky conditions due to the uncertainties in the model and observation operators under cloudy or rainy conditions. All experiments were conducted over a ten-day period from 13 to 22 October 2023. Among the available GMWR observations from August to October 2023, this period exhibited a notably higher frequency of

clear-sky data, which was more favorable for demonstrating the role and potential of GMWR assimilation. Before implementing bias correction, clear-sky screening, first-guess departure check, and whitelist check were sequentially applied to improve measurement quality. Subsequently, a relative departure check was applied prior to minimization. For the 6 h, 3 h, and 1 h assimilation intervals, 34 (0.91 %), 70 (1.42 %), and 76 (0.72 %) observations were rejected, respectively. Although the experiment with a 1 h assimilation interval rejected the largest number of observations, its rejection rate remains the lowest because it has the highest assimilation frequency and assimilates the largest volume of GMWR data. The detailed procedure prior to a single assimilation cycle is as follows:

(1) Observation Selection: The observation nearest to the analysis time within ±10 minutes is selected.

(2) Clear-sky Screening: Clear-sky GMWR observations are screened using the AGRI-based CLM, with background-simulated cloud liquid water path equal to zero.

(3) First-Guess Departure Check: Observations with |O−B| values greater than 20 K are excluded.

(4) Whitelist Check: Observations from stations identified as unreliable or displaying abnormal behavior are removed.

(5) Bias Correction: A machine learning bias correction scheme is applied (see Sect. 3.2).

(6) Relative Departure Check: This check is applied when the absolute value of the O−B exceeds three times the standard deviation of the observational error, further rejecting questionable data.

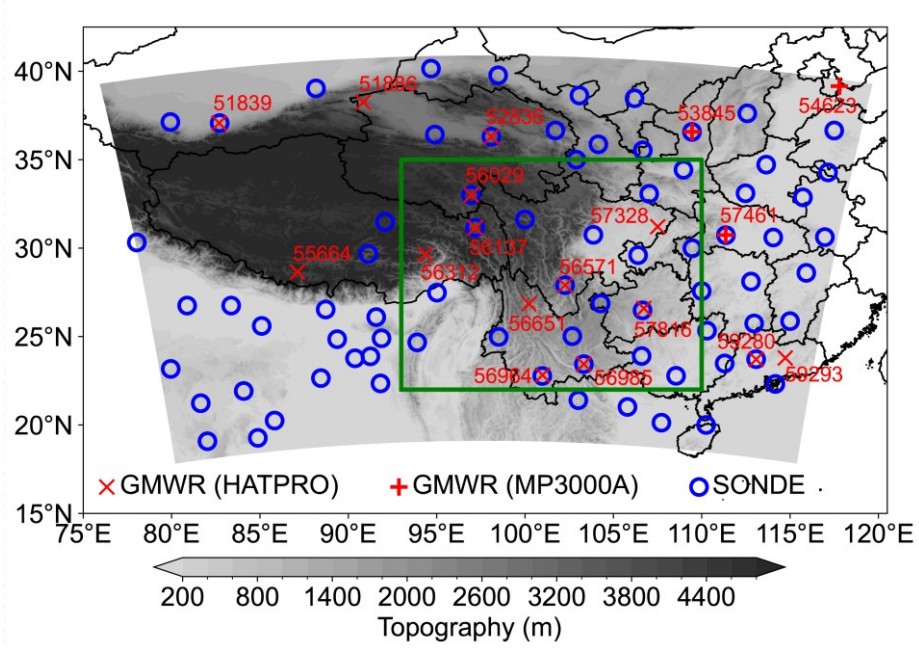

**Figure 1. Computational domain (shaded). The shading denotes topography (units: m). The green rectangle denotes the target region of Southwest China. The blue open circles denote radiosondes. The 'x' and '+' symbols denote HATPRO and MP3000A, respectively.**

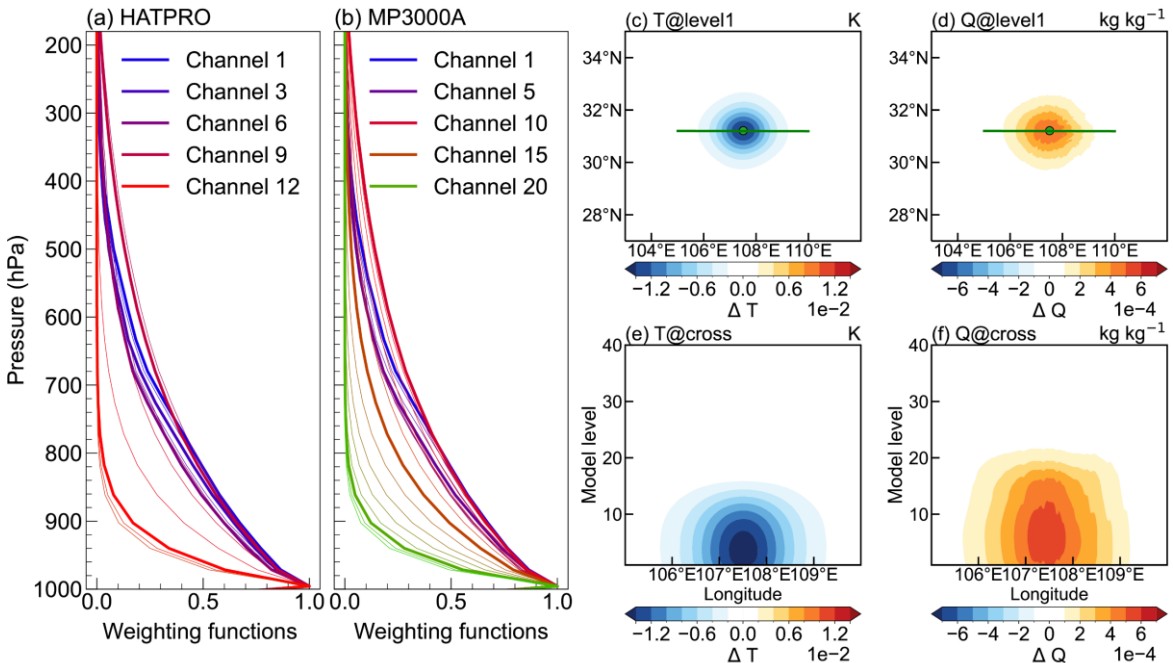

**Figure 2. Normalized weighting functions of (a) HATPRO and (b) MP3000A calculated using RTTOV-gb. The (c, d) horizontal and (e, f) vertical analysis increments for (c, e) temperature and (d, f) water vapor mixing ratio in single-observation assimilation experiment. The vertical increments are cross-sections along the green lines shown in the horizontal increments. The colorbar tick labels for temperature and water vapor mixing ratio are expressed in scientific notation as $1 \times 10^{-2}$ and $1 \times 10^{-4}$, respectively.**

Table 2 Experimental design

| Experiment | Assimilated Data | Assimilation Interval |
|---|---|---|
| CNTL | GTS | 6 h |
| GMWR_6H | GTS and GMWR | 6 h |
| GMWR_3H | GTS and GMWR | 3 h |
| GMWR_1H | GTS and GMWR | 1 h |

## 3 Machine learning based bias correction for GMWR

### 3.1 Bias characteristics

Variational assimilation assumes that both observation and background errors follow an unbiased Gaussian distribution. However, due to instrument errors, limitations of the RTMs, and errors in the NWP model background, observed brightness temperatures (O) and simulated brightness temperatures (B) inherently contain errors (denoted as $\mu^o$ and $\mu^b$), which may exhibit a biased distribution. Bias correction is a crucial process in radiance data assimilation, aiming to identify and remove

these biases (Auligné et al., 2007; Dee, 2005). In the real atmosphere, O and B are regarded as the true value (T) plus their respective deviations μ, as shown in Eq. (2):

$$\overline{O - B} = \overline{(O - T) - (B - T)} = \overline{\mu^o - \mu^b} \, , \tag{2}$$

It shows that the statistical expectation value of O−B can represent the systematic deviation ($\mu^o - \mu^b$). Therefore, it is critical to evaluate the bias characteristics of O−B and correct them.

To estimate the bias and develop a bias correction scheme for GMWR direct assimilation, a long-term experiment was conducted from August to October 2023, yielding a three-month sample dataset. In this experiment, the WRF model was initialized every 6 h using NCEP FNL data, and WRFDA operated hourly in monitoring mode (only calculating O−B). After a cloud check using the AGRI-based CLM and a gross check (|O−B| < 20 K), the bias of O−B for HATPRO and MP3000A was estimated.

A comparative scatterplot analysis of observed and simulated brightness temperatures was conducted. For most channels, the scatter points are closely aligned along the diagonal and exhibit high correlation coefficients, indicating strong agreement between the simulations and observations. However, the scatter points for some channels form two distinct clusters. To further investigate, representative channels from the K-band (water vapor absorption lines) and the V-band (temperature-sensitive oxygen absorption lines) were selected. Figure 3 presents scatterplots for channel 1 (K-band) and channel 13 (V-band) of HATPRO, as well as channel 1 (K-band) and channel 14 (V-band) of MP3000A. Results for the remaining channels are shown in Fig. A1 and Fig. A2. For HATPRO, more than 6,000 samples were analyzed. The O−B biases were 1.25 K for channel 1 and 2.14 K for channel 13, with standard deviations (STD) of 3.35 K and 2.82 K, respectively. Additionally, the scatter distribution for channel 13 is not centered, showing a cluster shifted to the right of the diagonal (Fig. 3b). For MP3000A, more than 2,000 samples were analyzed, with O−B biases of 3.06 K for channel 1 and −0.54 K for channel 14. The O−B STDs were 3.94 K and 3.08 K, respectively. Similar to the results for HATPRO channel 13 (V-band), the scatter points for MP3000A channel 14 (V-band) also show a cluster offset from the diagonal, but to the left (Fig. 3d). Based on these results, significant O−B biases are detected in GMWR observations, with their characteristics varying across different sensors and channels. However, the correlation coefficients between observed and simulated brightness temperatures are high, at least 0.95, suggesting that these biases can be effectively corrected.

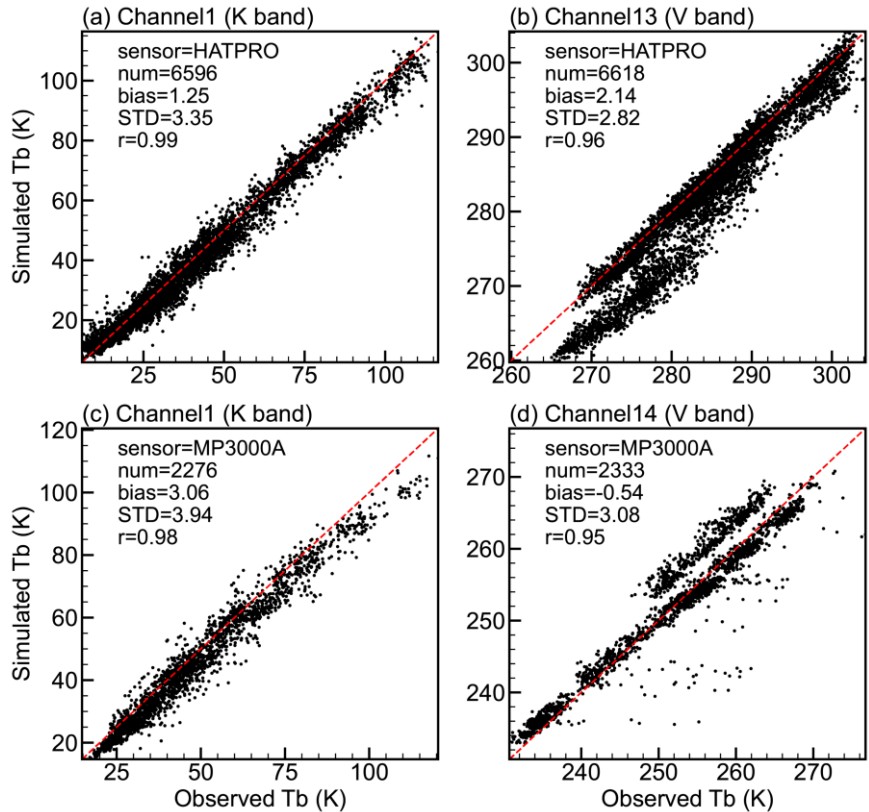

**Figure 3. Scatterplots of observed brightness temperature (Tb) versus simulated Tb, based on samples collected from August to October 2023. The top and bottom rows correspond to the HATPRO and MP3000A sensors, and the left and right columns represent the K-band and V-band, respectively. Each panel displays the number of samples (num), the O−B mean (bias), O−B standard deviation (STD), and the correlation coefficient (r) between observed and simulated Tb.**

To further analyze the O−B bias characteristics at each station and investigate the reasons for the band shifting from the diagonal (Fig. 3b and 3d), the statistics for each station are presented in Fig. 4. For HATPRO, the O−B bias varies among stations. Stations near complex topography (e.g., 56312, 56137, 56029, and 55664) exhibit notable positive O−B biases in channels 8–13 (Fig. 4a), leading to a rightward shift of the band relative to the diagonal (Fig. 3b). These positive biases may result from biases in the background field over the topographic region, the limited applicability of RTTOV-gb coefficients, or calibration issues in the observations. Overall, the O−B STD at each station for the K-band is larger than that for the V-band (Fig. 4b). The correlation coefficients between observed and simulated brightness temperatures are high across all channels (typically above 0.90), although they are slightly lower for channels 4–9. For MP3000A, station 57461 exhibits a negative O−B bias in channels 9–14 (Fig. 4d), contributing to the band shifting to the left of the diagonal (Fig. 3d). Similar to the results for HATPRO, the O−B STD in the K-band is generally larger than that in the V-band (Fig. 4e), and the correlation coefficients are also overall higher, typically exceeding 0.90 (Fig. 4f).

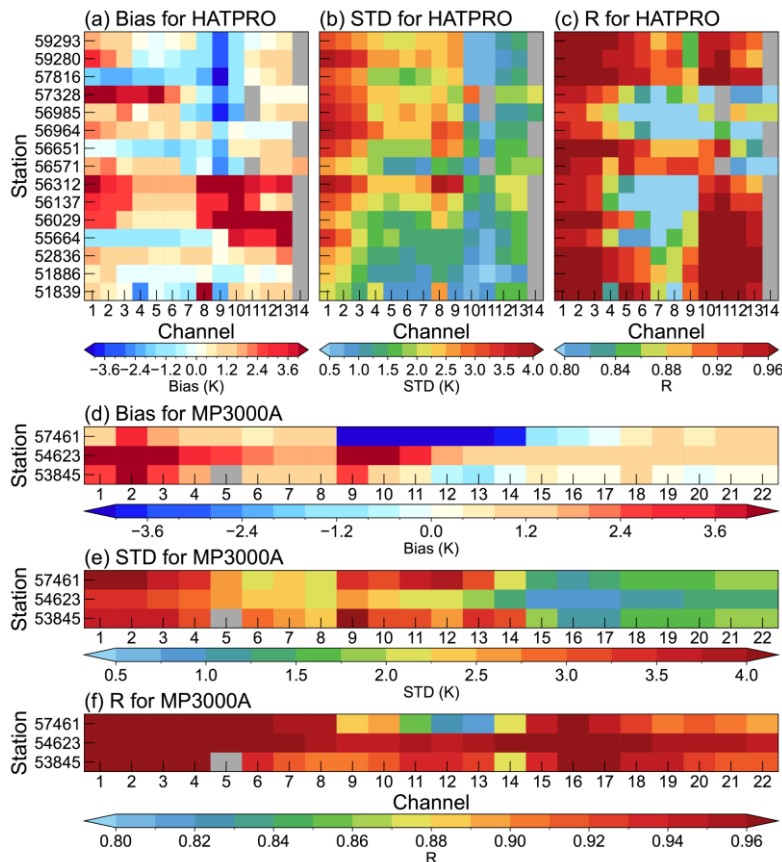

Figure 4. Statistics at each station based on samples collected from August to October 2023. O−B (a) bias and (b) standard deviation (STD) for HATPRO; (c) correlation coefficient (r) between observed and simulated brightness temperatures for HATPRO; (d–f) same as (a–c) but for MP3000A. Some stations did not provide observations for specific channels; the corresponding missing data are displayed in grey in the figure.

## 3.2 Bias correction

Based on the results above, noticeable O−B biases were observed, varying across sensors, channels, and the geographical locations of stations. It is essential to remove these biases before assimilation. Static bias correction (Harris and Kelly, 2001) and variational bias correction (Dee, 2005) are commonly used in radiance data assimilation. O−B bias is commonly represented using multiple linear regression with several predictors. Compared to linear estimates, nonlinear approaches show improved performance in reducing systematic biases (Zhang et al., 2023, 2024). Following these works, this study employed a machine learning–based bias correction scheme using the Random Forest (RF) technique (Breiman, 2001).

According to Yin et al. (2020), the predictors include the 1000–300 hPa thickness, 200–50 hPa thickness, model surface skin temperature (TS) and total precipitable water (PW). Considering that GMWRs are sensitive to the lower atmosphere, the predictors also include 1000–700 hPa thickness, 700–500 hPa thickness, 500–300 hPa thickness, 2 m temperature (T2), 2 m water vapor mixing ratio (Q2), 10 m zonal wind (U10), 10 m meridional wind (V10), and surface pressure (PS). Finally,

latitude, longitude, and observed brightness temperatures (Tb) are included as predictors due to their potential importance (Zhang et al., 2023). The O−B biases vary across sensors and channels. Therefore, a separate model is trained for each sensor and channel. Biases also vary across the geographical locations of stations, potentially influenced by the large-scale topography of the Tibetan Plateau. As predictors, 2 m temperature, surface pressure, latitude and longitude are important for explaining these biases.

There are two types of parameters in machine learning models: model parameters and hyperparameters. Model parameters are initialized and updated during the learning process. Hyperparameters, on the other hand, cannot be directly estimated from data. They must be configured before training because they define the model's architecture. Building an optimal machine learning model requires exploring a range of possibilities. The process of determining the ideal model architecture and hyperparameter configuration is known as hyperparameter tuning, which is a key component of developing an effective machine learning model (Yang and Shami, 2020).

The RF model has four key hyperparameters: the number of trees in the forest (*n_estimators*), the maximum depth of a tree (*max_depth*), the minimum number of samples required to split an internal node (*min_samples_split*), and the minimum number of samples required to be at a leaf node (*min_samples_leaf*). These hyperparameters were tuned using scikit-learn's GridSearchCV (Pedregosa et al., 2011) with 5-fold cross-validation, which exhaustively searches over a predefined range of hyperparameters, training and evaluating the model for each configuration. The flowchart illustrating the training and evaluation process of the bias correction (BC) model is shown in Fig. 5. The three-month sample dataset (described in Sect. 3.1) was randomly split into a training set (70 %) and a test set (30 %). During training, GridSearchCV constructed a large grid of possible hyperparameter configurations, iteratively trained and evaluated the model for each, and calculated a score. Finally, the optimized model was trained using the configuration with the highest score.

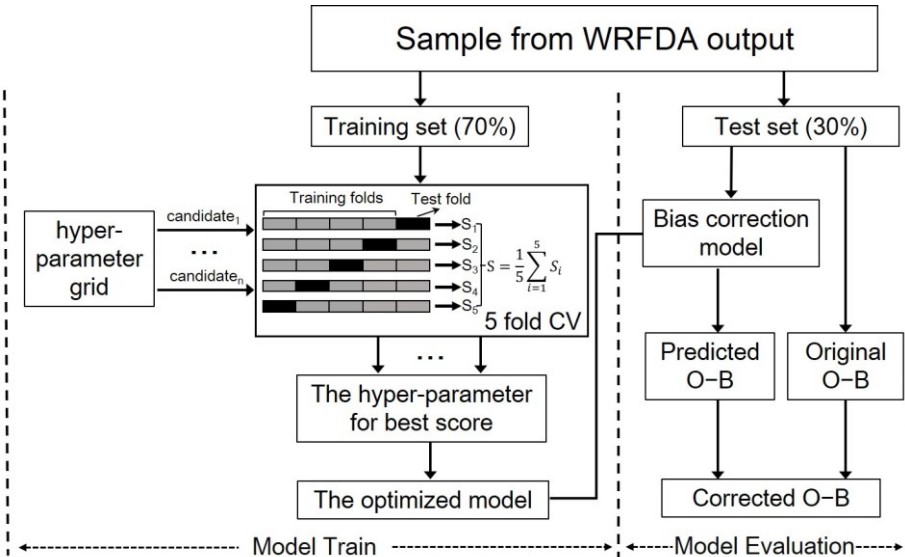

**Figure 5. Flowchart of the training and evaluation of bias correction model.**

To investigate the impact of hyperparameters on model training time and performance, the fit time and score of the RF model under various hyperparameter settings were analyzed (figure not shown). During hyperparameter tuning, *n_estimators* was varied between 10 and 150, *max_depth* was adjusted from 5 to 30, *min_samples_leaf* was tested with values between 1 and 3, and *min_samples_split* was tuned in the range of 2 to 6. Overall, the fit time and score tended to increase together. As the *min_samples_leaf* and *min_samples_split* parameters increased, both fit time and score decreased monotonically. Conversely, increasing the *max_depth* and *n_estimators* parameters resulted in a monotonic increase in both fit time and score. Notably, *max_depth* had the most significant impact on the score, while *n_estimators* primarily affected the fit time. For *n_estimators*, the score increased logarithmically, while the fit time grew linearly. These findings suggest that selecting a moderately small value for the *n_estimators* parameter can achieve better results while reducing computational time.

Using the above bias correction (BC) model, the corrected O−B values are obtained by subtracting the predicted O−B bias from the original O−B values. The effectiveness of the BC model is assessed based on the probability density functions (PDFs) of the O−B distributions in the test set (see Fig. 6 and Appendix B). Similar to Fig. 3, channel 1 (K-band) and channel 13 (V-band) of HATPRO, as well as channel 1 (K-band) and channel 14 (V-band) of MP3000A, are selected for detailed analysis (Fig. 6). For HATPRO channels 1 and 13, the biases (STDs) are 1.24 K (3.38 K) and 2.21 K (2.90 K), respectively. For MP3000A channels 1 and 14, the biases (STDs) are 3.00 K (3.89 K) and –0.64 K (3.08 K), respectively. The differences between the test set (Fig. 6) and the full dataset (Fig. 3) are negligible, with maximum differences in bias and STD of 0.10 K and 0.08 K, respectively, highlighting the strong representativeness of the test set. The original PDFs generally exhibit a unimodal shape, although their peaks deviate from zero. Moreover, some channels display bimodal features, often manifested as secondary peaks superimposed on the primary distribution—an issue that may affect 3DVAR, which typically assumes the errors to follow an unbiased Gaussian distribution.

After applying the BC model, both the O−B bias and STD are reduced, and the distribution becomes more sharply concentrated around zero, accompanied by an increase in kurtosis. For instance, in HATPRO channel 1, the bias (STD) decreases from 1.24 K (3.38 K) to 0.03 K (1.44 K), respectively, while the kurtosis markedly increases from 1.53 to 9.44. It is noted that the PDFs of corrected O−B approximate an unbiased distribution, and the secondary peaks are effectively suppressed. These results confirm that the proposed BC scheme effectively reduces both the bias and STD in the O−B statistics, transforming the PDFs from a bimodal to a unimodal distribution. The bimodal feature in the O−B PDFs corresponds to the two distinct clusters observed in the scatter plots shown in Fig. 3. Specifically, when one cluster is concentrated along the diagonal and the other shifts to the right, a peak forms on the positive x-axis of the O−B PDFs, resulting in a bimodal distribution. From the O−B PDFs (Fig. 6), both instruments exhibit a positive bias with a unimodal distribution in the K-band. In contrast, the V-band displays a bimodal distribution: the second peak appears on the right for HATPRO and on the left for MP3000A. These results are consistent with the scatter plots shown in Fig. 3. After BC, the O−B PDFs change from a bimodal to a unimodal distribution, indicating that the two clusters have been effectively merged.

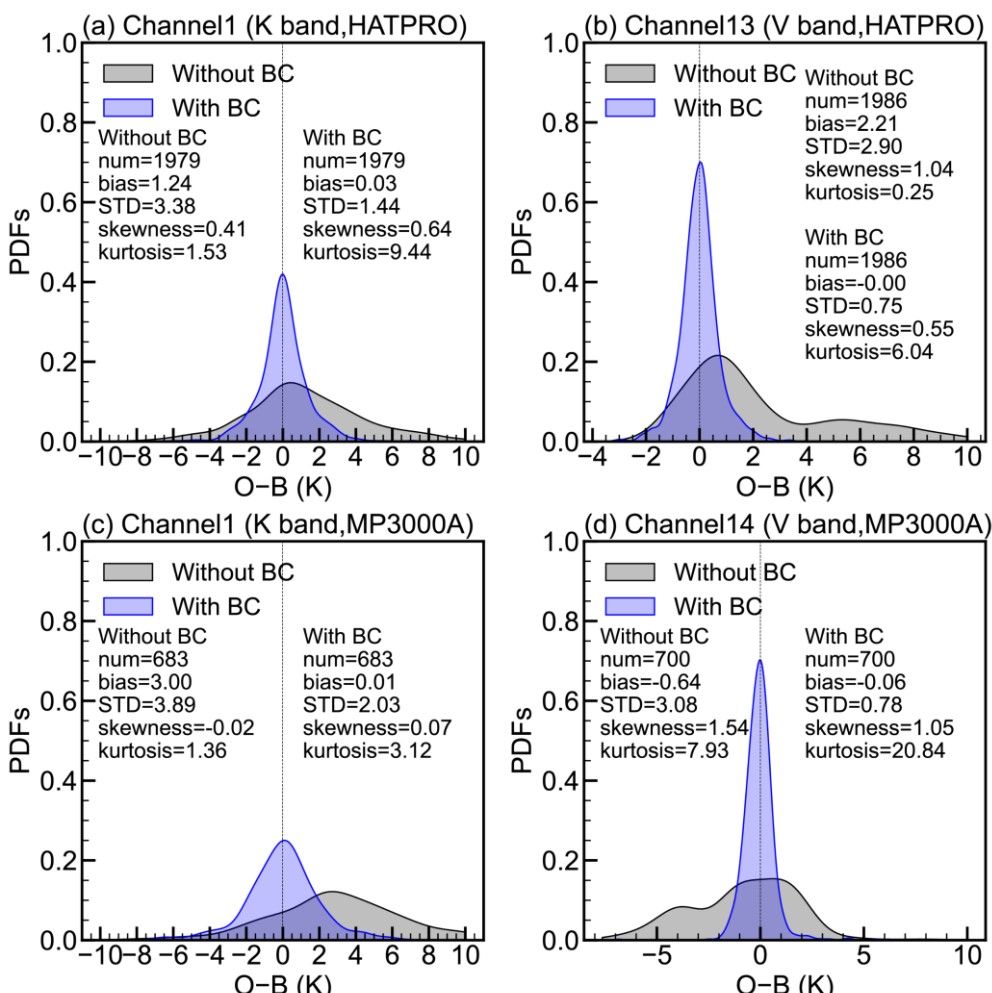

Figure 6. Probability density functions (PDFs) of the O−B distributions, based on a test set randomly selected from 30 % of the three-month sample dataset collected from August to October 2023. The top and bottom rows correspond to the HATPRO and MP3000A sensors, respectively, while the left and right columns represent the K-band and V-band, respectively. Each panel displays the number of samples (num), the mean (bias), standard deviation (STD), skewness, and kurtosis of the distributions.

Figure 7 illustrates the O−B bias and STD for each HATPRO channel. Before BC, the O−B bias ranged from 0 to 2 K, with the bias in the K-band (particularly channels 4–7) smaller than that in the V-band. After BC, the bias for each channel is approximately 0 K. In terms of the O−B STD, values ranged from 2 to 4 K without BC, with channels 4–7 exhibiting smaller values compared to other channels. After BC, the STD of O−B ranges between 0.5 and 1.5 K. The application of this BC model significantly reduced both the O−B bias and STD, with reductions of 0.83 K (97.1 %) and 1.63 K (64.6 %), respectively. Meanwhile, the corrected O−B distributions display approximately Gaussian characteristics centered around zero, indicating effective removal of systematic O−B biases.

Diagnosing the contributions of each predictor is crucial. Figure 7c illustrates the feature importance of several predictors for HATPRO. This BC model normalizes the feature importance scores so that their sum equals 1. A higher score reflects a stronger influence of the predictor on the O−B biases. Observed brightness temperature, total precipitable water, and surface pressure are significant contributors to BC for the K-band (water vapor channels). For the V-band (temperature channels), observed brightness temperature, latitude, and surface pressure are the most influential predictors. The contributions of atmospheric thickness predictors are smaller compared to the other predictors; however, the 1000–700 hPa thickness predictor has a relatively larger contribution among them. This may be because GMWR primarily observes radiation from the lower atmosphere. Notably, surface pressure plays a critical role in BC for the V-band, which may account for the positive bias in O−B observed at plateau stations (Fig. 4a).

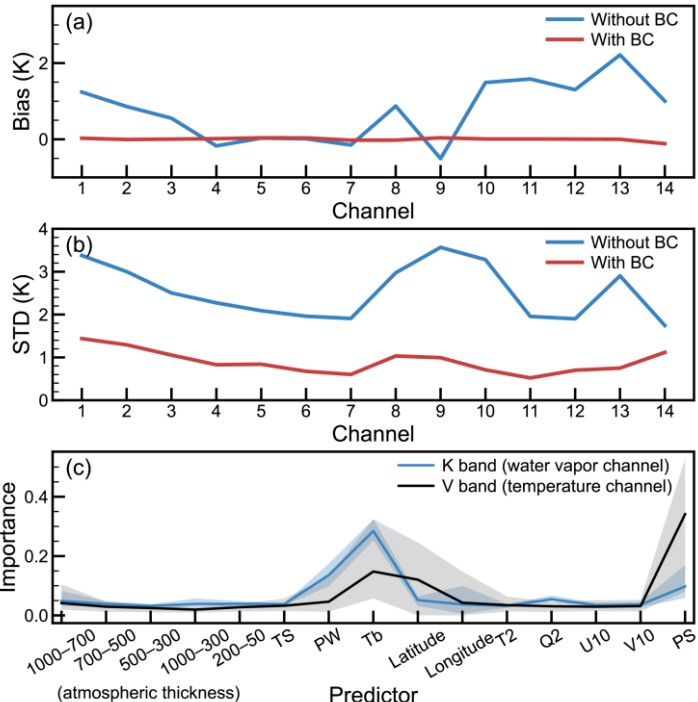

Figure 7. (a) Bias and (b) standard deviation (STD) of O−B, based on a test set randomly selected from 30 % of the three-month sample dataset collected from August to October 2023. (c) Feature importance of the predictors used in the bias correction (BC) model. For each band, the shaded regions and solid lines represent the range and mean feature importance, respectively.

## 4 Direct assimilation of GMWR radiance observations

### 4.1 Assimilation impacts on initial conditions

The performance of GMWR assimilation in the observation space was evaluated. Figure 8 summarizes the biases and STDs of the O−B and observation minus analysis (O−A) statistics, aggregated over time for different channels. The bias of O−A was reduced compared to O−B, particularly in the V-band. Specifically, for channel 11 of GMWR_1H (1 h assimilation interval),

O−B was −0.40 K and O−A was −0.13 K. When GMWR observations were assimilated, the simulated brightness temperatures became closer to the observations, resulting in smaller STDs. Moreover, as the frequency of GMWR observation assimilation increases, the bias and STD of the O−B gradually converge toward zero. For channel 3, the O−B STD in GMWR_6H, GMWR_3H, and GMWR_1H significantly decreased from 1.03 K, 0.92 K, and 0.56 K to O−A STD values of 0.36 K, 0.34 K, and 0.34 K, respectively. Although the differences in O−A among experiments are less noticeable, the improvement of O−B suggests that increasing the frequency in cycling assimilation accumulates the impact of the GMWRs, producing a higher-quality first-guess field for the final cycle. O−A statistics were also computed based on the initial fields from the CNTL experiment. The O−A bias in CNTL is slightly larger than that in the GMWR assimilation experiments, with a more noticeable difference in the V-band. Regarding the STD, the CNTL experiment shows higher values across all channels, with the largest difference approaching 1 K. These results suggest that assimilating GMWR data improves the consistency between the initial fields and the observed brightness temperatures. The assimilation of GMWR observations effectively influences the brightness temperatures, demonstrating the successful processing of GMWR data by the 3DVAR system.

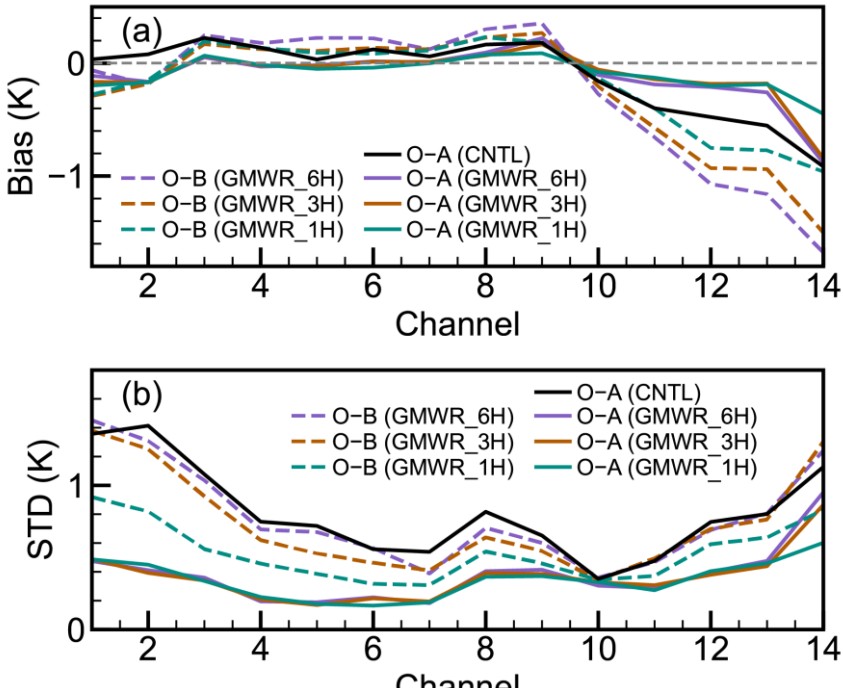

**Figure 8. Verification of the initial conditions against GMWR observations, based on the ten-day assimilation experiment conducted from 13 to 22 October 2023. (a) Bias and (b) standard deviation (STD) of the observation minus background (O−B) and observation minus analysis (O−A) for the GMWR assimilation in the target region of Southwest China (green rectangle in Fig. 1).**

The above evaluation demonstrates the successful implementation of the newly introduced GMWR radiance direct assimilation in WRFDA. However, compared to brightness temperature simulations, greater attention should be given to the model state variables in the initial conditions, as they directly influence subsequent model forecasts. To this end, radiosonde observations

in the target region of Southwest China were used to evaluate the impact of GMWR assimilation. The root-mean-square error

(RMSE) was calculated, and the RMSE differences between CNTL and other assimilation experiments are shown in Fig. 9. Results indicate that assimilating GMWR radiances enhances low-level temperature and humidity fields, with higher assimilation frequencies offering the potential for additional improvements. GMWR assimilation has a neutral impact on atmospheric temperature above 1 km above ground level (AGL), where the RMSE difference is minimal. However, it positively impacts lower atmospheric temperature, with the RMSE for temperature decreasing below 1 km AGL. Specifically,

the average RMSE improvements below 1 km are 3.67 %, 5.28 %, and 6.32 % for GMWR_6H, GMWR_3H, and GMWR_1H, respectively. This indicates that increasing assimilation frequency enhances observational impacts and further improves the initial conditions. The improvement becomes more pronounced with decreasing altitude. At 100 m AGL, RMSE reductions are 0.10 K (6.25 %), 0.13 K (7.90 %), and 0.19 K (11.34 %) in GMWR_6H, GMWR_3H, and GMWR_1H, respectively. For the water vapor mixing ratio (QVAPOR), GMWR assimilation demonstrates a positive impact extending into the middle

troposphere, with average RMSE reductions below 5 km of 2.30 %, 2.20 %, and 1.98 % for GMWR_6H, GMWR_3H, and GMWR_1H, respectively. The impact of GMWR assimilation and the effect of assimilation frequency are generally more pronounced in the lower atmosphere, with average RMSE reductions below 300 m of 3.01 % for GMWR_1H, compared to 2.43 % for GMWR_6H and 2.05 % for GMWR_3H.

It should be noted that the GMWR assimilation shows a slight degradation in the wind fields. The RMSE for zonal and

365 meridional winds exhibits a slight negative effect when GMWR is assimilated, with meridional winds even showing an increase in RMSE. These negative impacts on the wind field caused by GMWR assimilation may be attributed to two factors: (1) When assimilating observed brightness temperatures, the adjoint model of the observation operator directly adjusts temperature and humidity to optimize the simulation, while changes in the wind field are indirectly driven by these adjustments through the background error covariance. (2) GMWR assimilation primarily improves the lower atmosphere, while changes

in the upper atmosphere are also governed by the background error covariance. The static background error covariance used here is climatological and isotropic, which does not fully align with evolving weather conditions, potentially resulting in ineffective wind field improvements.

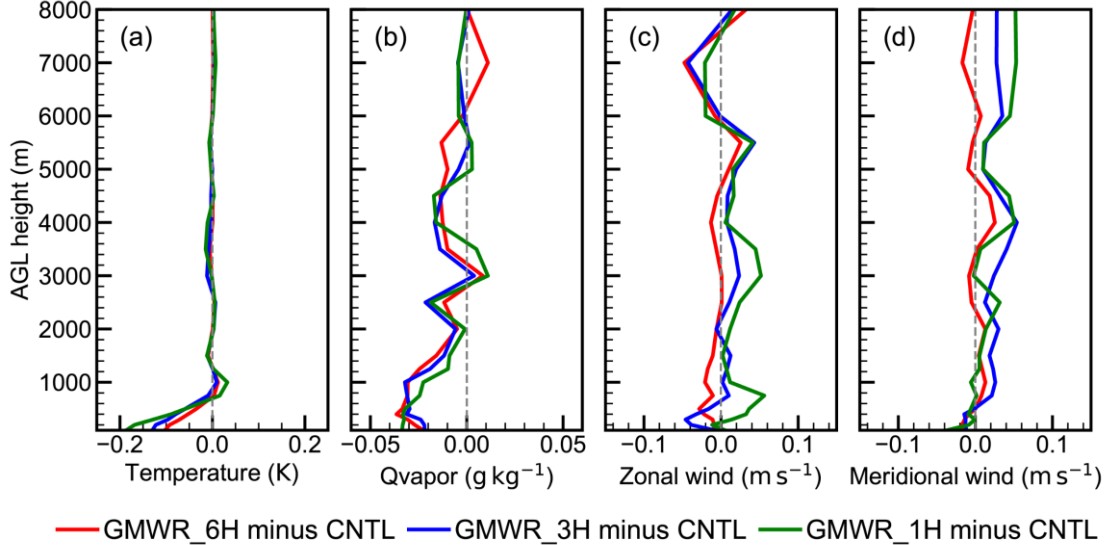

**Figure 9. Verification of the initial conditions against radiosonde observations. RMSEs are computed from all samples over the ten-day assimilation experiment conducted from 13 to 22 October 2023. Root mean square error (RMSE) of (a) temperature, (b) water vapor mixing ratio (QVAPOR), (c) zonal wind, and (d) meridional wind in the target region of Southwest China (green rectangle in Fig. 1).**

Based on the evaluation against radiosonde observations, the assimilation of GMWR data improves the initial fields of temperature and humidity, aligning them more closely with observations, particularly in the lower atmosphere. Additionally, the initial fields are validated against surface station observations, including measurements of 2 m temperature, 2 m relative humidity, and 10 m wind (Fig. 10). The RMSE differences indicate that GMWR assimilation effectively enhances the 2 m temperature and humidity fields. Under 6-hourly GMWR assimilation, the temperature RMSE generally increased on the southern side of the basin, whereas other regions showed a positive effect with reduced RMSE values. Moreover, the temperature RMSE reduction in these positively affected areas further improved as the assimilation frequency increased, with overall differences ranging from $-0.008$ K ($-0.3$ %) to $-0.099$ K ($-4.1$ %). For humidity, GMWR assimilation shows a negative impact on 2 m relative humidity (RH) at a 6 h assimilation frequency. However, the RMSE over the plateau decreases as the assimilation frequency increases, with the RMSE difference shifting from positive to negative. In the GMWR_1H experiment, the RMSE is reduced by 0.276 (1.3 %).

Unlike the temperature and humidity RMSEs, the improvement in the wind field RMSE does not exhibit a distinct spatial pattern. Compared to the CNTL experiment, the RMSE differences for zonal wind are $-0.005$ m s$^{-1}$ ($-0.3$ %), $-0.017$ m s$^{-1}$ ($-1.0$ %), and $-0.019$ m s$^{-1}$ ($-1.2$ %) in GMWR_6H, GMWR_3H, and GMWR_1H, respectively. Similarly, the RMSE differences for meridional wind are $-0.008$ m s$^{-1}$ ($-0.5$ %), $-0.011$ m s$^{-1}$ ($-0.7$ %), and $-0.009$ m s$^{-1}$ ($-0.5$ %) in GMWR_6H, GMWR_3H, and GMWR_1H, respectively. While the changes in wind RMSE are relatively small, the results indicate that assimilating GMWR data improves the initial conditions, with higher assimilation frequencies offering potential for further enhancement.

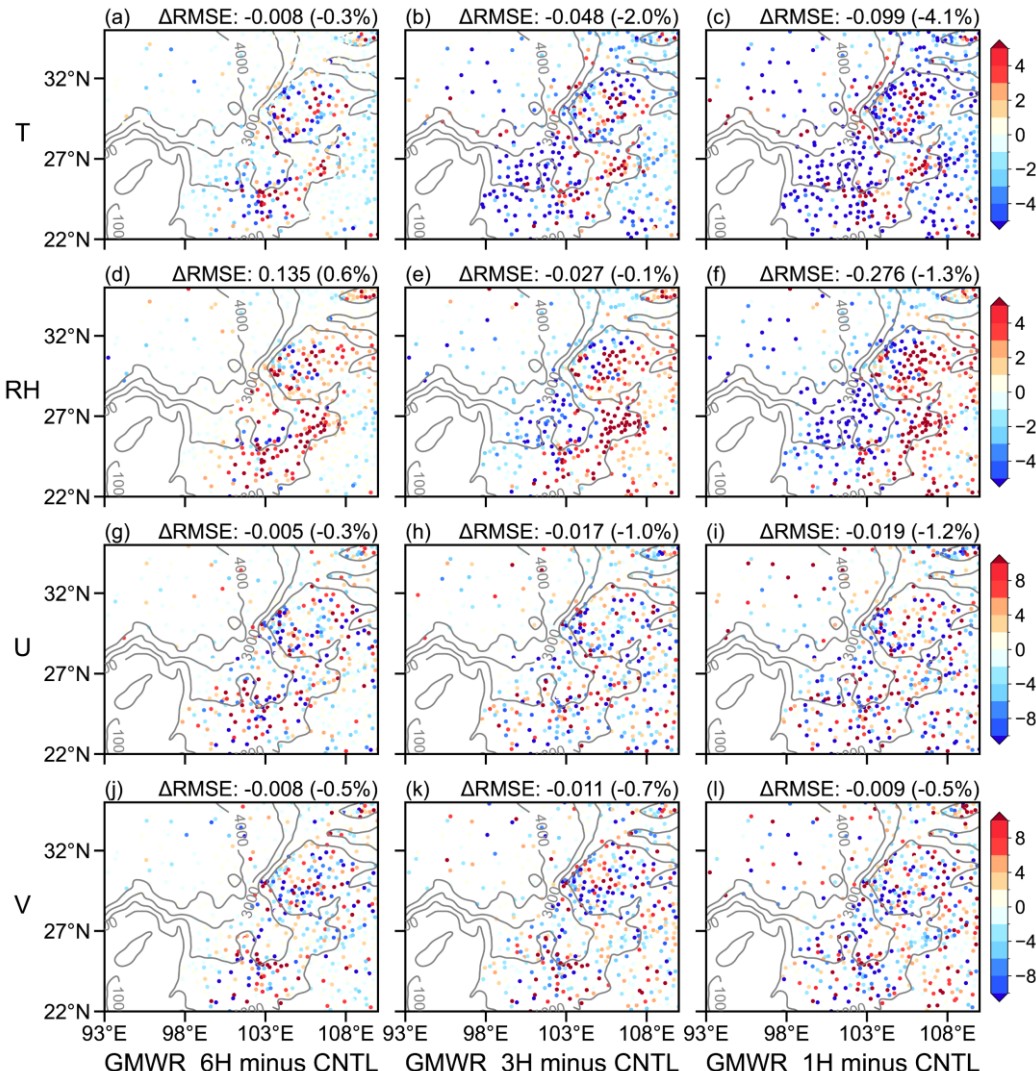

**Figure 10. Verification of the initial conditions against surface station observations, based on the ten-day assimilation experiment conducted from 13 to 22 October 2023. The percentage differences in RMSE (scatter) of temperature (T), relative humidity (RH), zonal wind (U), and meridional wind (V) in the target region of Southwest China (green rectangle in Fig. 1). The grey solid line represents the topography height (m).**

### 4.2 Assimilation impacts on forecast field

After presenting the improvements in the initial conditions, this section investigates the impact of GMWR assimilation on the 24 h forecasts. The time series of RMSE for the CNTL experiment and RMSE differences (assimilation experiments minus the CNTL experiment) against surface station observations for 2 m temperature, 2 m relative humidity, and 10 m wind fields are shown in Fig. 11. In the CNTL experiment, the RMSE of temperature and relative humidity initially decreases and then increases with lead time, while the RMSE of the wind field exhibits the opposite trend, increasing at first and then decreasing.

The mean RMSEs over the 24 h forecast period are 2.32 K for temperature, 16.26 % for relative humidity, 1.92 m s⁻¹ for zonal wind, and 2.08 m s⁻¹ for meridional wind. Regarding assimilation impacts, the RMSE reduction for temperature gradually decreases, approaching zero at a lead time of 6 h, with higher assimilation frequency (GMWR_1H) achieving a greater RMSE reduction. Similar results are observed for relative humidity, where the RMSE reduction also decreases and approaches zero at a lead time of 12 h. GMWR_1H consistently demonstrates the largest RMSE reduction for relative humidity. However, it should be noted that the direct assimilation of GMWR data caused a negative impact on relative humidity at a lead time of 12 h. The degradation of wind fields (Fig. 9) and the model's inherent nonlinearity may be responsible. For the wind field, no increase in the RMSE difference with lead time was observed, as previously described. However, the RMSE differences between the assimilation experiments and the CNTL experiment remain overall negative, indicating that GMWR assimilation improves wind forecasts. Additionally, GMWR_1H demonstrates the largest RMSE reduction in meridional wind, suggesting that increasing the frequency of GMWR assimilation may lead to further improvements. The quantitative statistics are presented in Table 3. The temperature RMSE differences between GMWR_6H and CNTL are −0.012, −0.005, and −0.004 K for lead times of 1–6 h, 1–12 h, and 1–24 h, respectively. This gradual decrease in RMSE differences with increasing forecast time is also observed in other experiments and variables, indicating a weakening of the positive impact of GMWR assimilation as the forecast period extends. When the impact of GMWR assimilation is most pronounced (at a lead times of 1–6 h), the temperature RMSE differences range from −0.012 K in GMWR_6H to −0.014 K in GMWR_3H, and −0.019 K in GMWR_1H. The temperature RMSE reduction increases with the frequency of GMWR assimilation, a trend also observed in relative humidity and wind, suggesting that increasing the assimilation frequency can further improve the short-term forecasts. Although these differences are small, the results reflect the potential for improved model forecasts with GMWR assimilation.

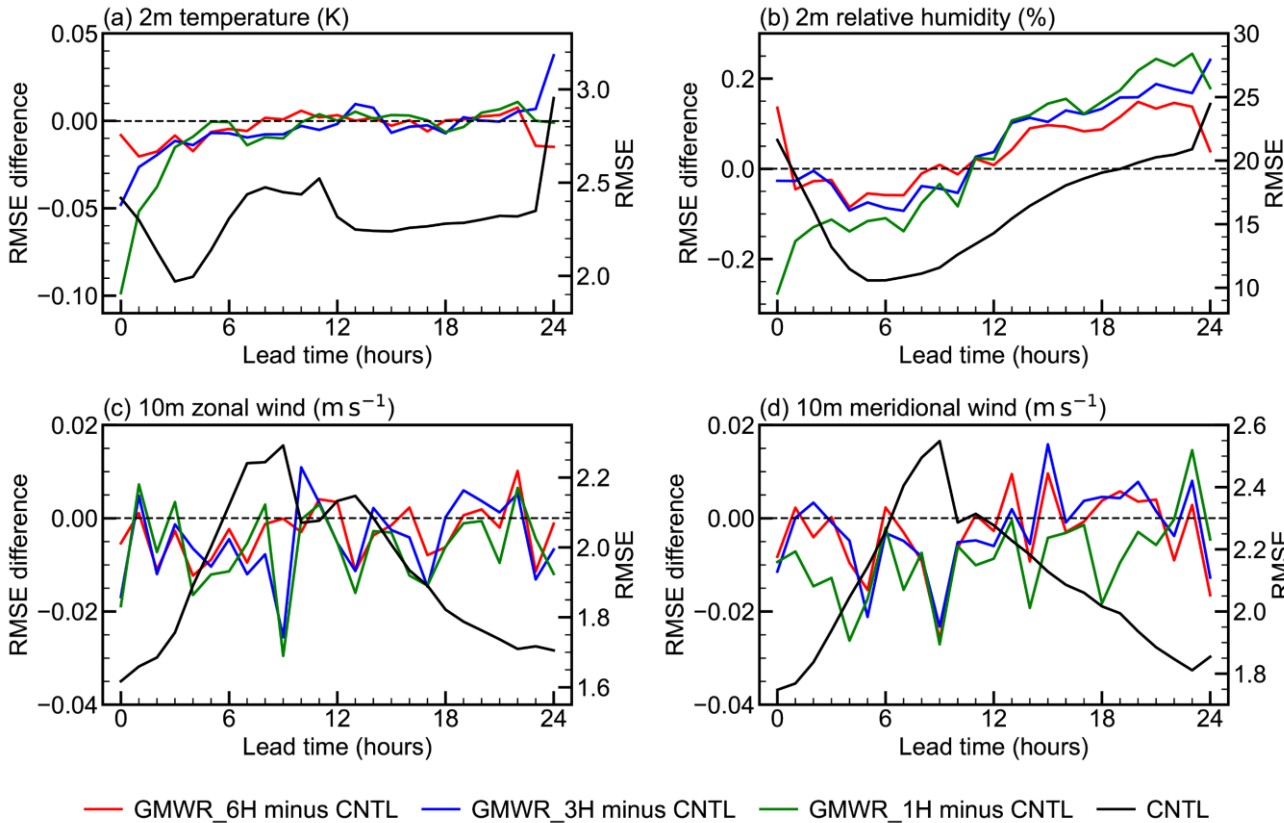

**Figure 11. Verification of the forecasts against surface station observations, based on the ten-day assimilation experiment conducted from 13 to 22 October 2023. RMSE (black line) for the CNTL experiment and RMSE differences (colored lines) between the assimilation experiments and the CNTL experiment for (a) temperature, (b) relative humidity, (c) zonal wind, and (d) meridional wind.**

Table 3 RMSE difference against surface station observations.

| EXP | Lead time (hour) | Temperature (K) | Relative Humidity (%) | Zonal wind (m s⁻¹) | Meridional wind (m s⁻¹) |
|---|---|---|---|---|---|
| GMWR_6H minus CNTL | 1–6 | −0.012 | −0.046 | −0.006 | −0.004 |
| | 1–12 | −0.005 | −0.027 | −0.003 | −0.006 |
| | 1–24 | −0.004 | 0.046 | −0.003 | −0.003 |
| | 1–6 | −0.014 | −0.046 | −0.005 | −0.005 |

| GMWR_3H minus CNTL | 1–12 | −0.010 | −0.036 | −0.006 | −0.007 |
|---|---|---|---|---|---|
| | 1–24 | −0.003 | 0.072 | −0.005 | −0.003 |
| GMWR_1H minus CNTL | 1–6 | −0.019 | −0.127 | −0.007 | −0.013 |
| | 1–12 | −0.012 | −0.087 | −0.006 | −0.013 |
| | 1–24 | −0.005 | 0.065 | −0.006 | −0.009 |

Verification against surface station observations indicated that assimilating GMWR radiances improves near-surface forecasts, with higher assimilation frequencies offering potential for further enhancement. To further examine the impact of GMWR assimilation, Fig. 12 presents the forecast verification against radiosonde observations. Unlike the RMSE differences in the initial conditions (Fig. 9), the GMWR assimilation did not reduce RMSE for temperature and water vapor mixing ratio, indicating a neutral impact on forecasts. Similarly, the wind field verification results did not show noticeable improvements

with GMWR assimilation. While the RMSE of zonal wind was reduced in the GMWR_1H experiment, the RMSE differences for the wind field in other experiments were close to or greater than zero, suggesting a neutral to slightly negative impact of GMWR assimilation on wind forecasts. Based on radiosonde-based verification at 12 and 24 h lead times, only limited improvements are evident after GMWR assimilation; however, this does not rule out larger impacts within 0–6 h, which cannot be robustly assessed here due to the limited temporal availability of radiosonde observations. The limited improvement shown

in this figure could be related to the relatively long forecast lead times (12 and 24 h), during which model errors tend to accumulate and weaken the benefits of improved initial conditions from GMWR assimilation. In the surface-station verification, the improvements were primarily confined to the first few hours, particularly for temperature and humidity. After 12 h, the impact declined noticeably, with some cases even exhibiting negative effects (Fig. 11).

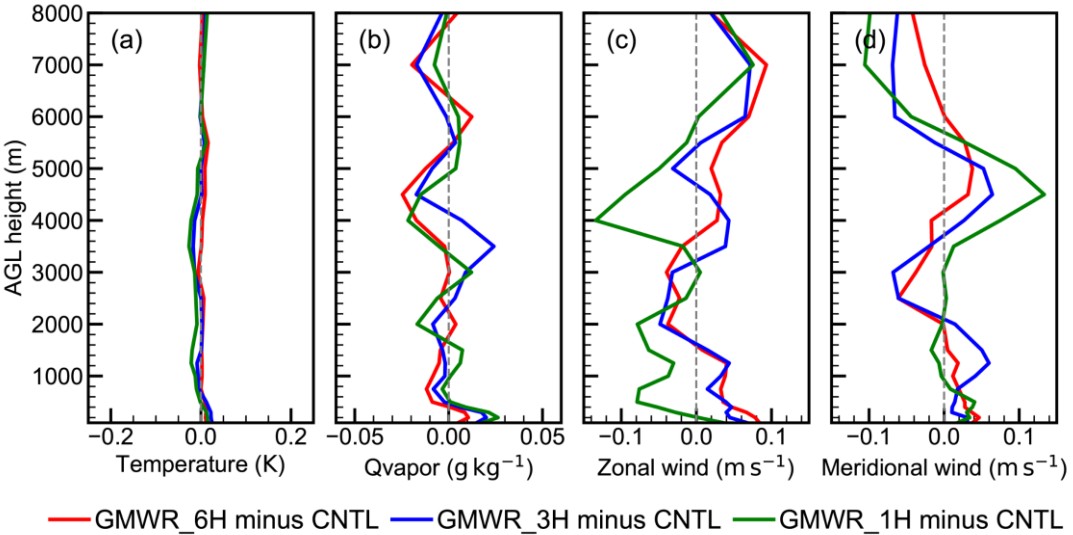

 **Figure 12. Same as Fig. 9, but for forecasts at lead times of 12 and 24 h, with RMSEs computed from all forecast samples during the ten-day experiment.**

To further explore the role of GMWR assimilation in precipitation forecasting, the fractions skill score (FSS) of 3 h accumulated precipitation forecasts was calculated. The radius of influence for the FSS was set to 18 km, equivalent to six times the grid spacing (Ha and Snyder, 2014; Zheng et al., 2024). Figure 13 presents the time series of FSS for the CNTL experiment and FSS differences (assimilation experiments minus the CNTL experiment). The assimilation experiments were conducted during a period with a high frequency of clear-sky observations. Cloud cover and precipitation were sparse over the ten-day period, resulting in the absence of frequent heavy rainfall events. Consequently, the FSS was calculated using small precipitation thresholds. In the CNTL experiment, the FSS for 3 h accumulated precipitation showed an initial decline followed by a subsequent increase with lead time, with relatively low FSS values observed around the 12 h forecast period. Moreover, the FSS generally decreases as the precipitation threshold increases. The time mean FSS values are 0.47, 0.45, 0.42, and 0.39 for thresholds of 3 mm, 4 mm, 5 mm, and 6 mm, respectively. Regarding the role of GMWR assimilation in precipitation forecasting, the results indicate that assimilating GMWR radiances enhances precipitation forecasts, with FSS differences increasing progressively at higher precipitation thresholds. Additionally, increasing assimilation frequency showed the potential to further enhance forecast performance. When assimilating GMWR data at a 1 h frequency, the time-averaged FSS improvements for 3 h accumulated precipitation are 0.02 (3.9 %) for the 3 mm threshold, 0.02 (4.7 %) for the 4 mm threshold, 0.03 (7.3 %) for the 5 mm threshold, and 0.04 (10.2 %) for the 6 mm threshold precipitation. For 3 h accumulated precipitation with a threshold of 6 mm, the time-averaged FSS improvements are 0.01, 0.02, and 0.03 for GMWR_6H, GMWR_3H, and GMWR_1H, respectively. These findings are consistent with the above verification against surface station observations,

suggesting that GMWR assimilation can improve forecasts and that higher-frequency assimilation leads to further enhancements.

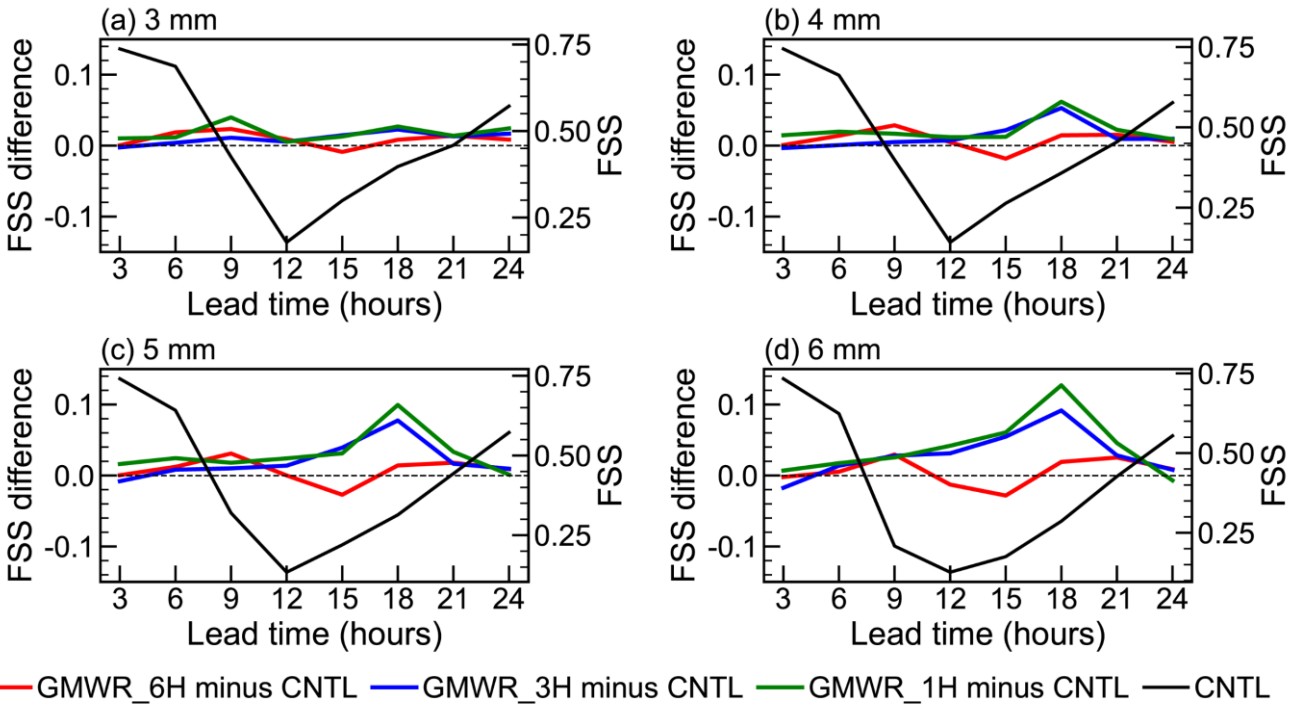

**Figure 13. The time series of FSS (black line) for the CNTL experiment and FSS differences (colored lines) between the assimilation experiments and the CNTL experiment. These experiments were conducted from 13 to 22 October 2023. The FSS was calculated for**
**3 h accumulated precipitation with thresholds of (a) 3 mm, (b) 4 mm, (c) 5 mm, and (d) 6 mm.**

## 5 Discussion, conclusions, and future work

### 5.1 Discussion

This section discusses (i) O−B characteristics and their potential sources, (ii) the effectiveness and physical interpretability of the RF-based bias correction (BC), and (iii) the impacts of direct GMWR radiance assimilation on analyses and forecasts.

In the three-month O−B statistics, the STD in the K-band is larger than that in the V-band, consistent with Vural et al. (2024) and Cao et al. (2023). A notable positive O−B bias is observed at high-altitude stations over the Tibetan Plateau, which may be related to large-scale topographic effects. In this region, model simulations may contain errors, and RTTOV-gb coefficients may be inapplicable. The RTTOV-gb coefficients are based on global atmospheric profiles, which may differ significantly from the climatic conditions of plateau regions, potentially affecting simulation accuracy.

To mitigate these systematic O−B biases, a machine learning-based BC scheme using the RF technique was developed. The number and depth of trees are critical hyperparameters that must be predetermined. Training time increases approximately linearly with the number of trees, while performance exhibits a logarithmic-like saturation trend. In terms of tree depth, both

training time and performance increase approximately logarithmically with depth. Thus, selecting a modest number ($n\_estimators$) and depth ($max\_depth$) of trees, such as 50 and 15, can balance efficiency and accuracy. Feature importance analysis for BC predictors revealed observed brightness temperature, atmospheric precipitable water, and surface pressure as key factors for correcting biases. The importance of brightness temperatures aligns with findings in satellite data bias correction (Liu et al., 2022; Zhang et al., 2023). Atmospheric precipitable water is essential for the K-band, a humidity-sensitive channel. Surface pressure plays a key role in temperature channels, thereby accounting for the positive bias observed in plateau regions. Although atmospheric thickness predictors contributed less overall, the 1000–700 hPa thickness was relatively significant, likely due to GMWRs primarily sensing radiation from the lower atmosphere.

The machine learning-based BC scheme effectively mitigated the bimodal distribution and systematic errors in the O−B statistics. To assess its impact on the initial and forecast fields, a parallel experiment without bias correction was conducted, based on the 1 h assimilation interval experiment (GMWR_1H). For the initial fields, as verified against radiosonde observations, the experiment without BC yielded only minor improvements in temperature and even degraded the water vapor field. As for the forecast fields, verification against surface station observations showed that the absence of BC led to a noticeable degradation in the forecast accuracy of 2 m temperature and relative humidity. These findings indicate that the machine learning-based BC scheme had a beneficial impact on both the initial conditions and the subsequent forecasts. Nevertheless, despite its demonstrated effectiveness, the scheme is subject to several limitations. Relying on offline O−B statistics, it implicitly assumes that all biases originate from the observations—an assumption that may not always hold and may, in some instances, mask model biases (Auligné et al., 2007; Eyre, 2016). These limitations motivate further improvements in future work (Sect. 5.2).

In this study, direct assimilation of GMWR radiances enhances both the initial conditions and the forecasts, showing potential for improving ABL and precipitation simulations. Although the assimilation of GMWR radiances yields slight improvements in the forecast wind fields (Fig. 11), it exerts an overall negative impact on the wind fields in the initial conditions (Fig. 9). It should be noted that assimilating GMWR radiances improves the wind fields below 500 m AGL in the initial conditions. This improvement is consistent with the verification of the forecast, which demonstrates enhancements in the 10 m wind fields. Regarding the degradation of wind fields above 500 m AGL, the background error covariance may contribute to this negative impact. Specifically, it propagates the RTTOV-gb increments concentrated in the ABL to higher levels and induces wind field adjustments in response to temperature and humidity updates.

## 5.2 Conclusions and future work

To investigate the impact of directly assimilating GMWRs in Southwest China, a GMWR assimilation module has been developed in WRFDA-4.5, where RTTOV-gb is used as the observation operator. Based on this module, a three-month sample dataset of O−B was collected to evaluate the bias and develop a BC model. Furthermore, ten-day assimilation experiments

(Table 2) were conducted using this GMWR assimilation module and BC model to investigate the impact of direct GMWR assimilation and the effects of assimilation frequency. The main findings are as follows:

(1) Based on three months of hourly samples, noticeable O−B biases were observed, varying across sensors, channels, and geographical locations. The machine learning-based bias correction scheme, employing an RF model, effectively reduced these O−B systematic biases. After applying this BC model, both the bias and STD of the O−B were substantially reduced. Specifically, the bias and STD decreased by 0.83 K (97.1 %) and 1.63 K (64.6 %), respectively. For some channels, the original O−B distribution exhibited a bimodal pattern, which was transformed into a unimodal distribution after BC. The corrected O−B distributions exhibited Gaussian characteristics centered around zero.

(2) Assimilating GMWR radiances enhances the accuracy of initial conditions, with higher assimilation frequencies amplifying the positive impact, particularly for temperature and humidity in the lower atmosphere. Evaluation against radiosonde observations shows that the temperature RMSE below 1 km AGL decreases by 3.67 % to 6.32 % as the assimilation frequency increases from 6 h to 1 h. For the water vapor mixing ratio, positive impacts extend up to 5 km AGL, with average RMSE improvements ranging from 1.98 % to 2.30 %. Verification against surface station observations further supports these findings, indicating that the RMSE for 2 m temperature decreases by up to 4.1 %, while the RMSE for 2 m relative humidity decreases by up to 1.3 % at the 1 h assimilation frequency.

(3) The assimilation of GMWR observations leads to improvements in forecasts, and increasing assimilation frequencies has the potential to yield further improvements. In the first 6 h of the forecast, the temperature RMSE decreases by 0.012 K, 0.014 K, and 0.019 K with 6 h, 3 h, and 1 h assimilation frequency, respectively. Similar trends are observed for relative humidity, where the experiment with 1 h GMWR assimilation frequency shows the largest decrease in RMSE. GMWR assimilation also improves precipitation forecasts, with further enhancements seen as assimilation frequency increases. For 1 h GMWR assimilation, time-averaged FSS improvements reach 0.02 for both the 3 mm and the 4 mm, 0.03 for the 5 mm, and 0.04 for the 6 mm thresholds.

Despite these encouraging results, this study has some limitations that motivate future work. Regarding bias correction, the offline scheme lacks anchoring observations, rendering the analysis fields more susceptible to model bias. Future efforts should consider bias correction strategies based on unbiased reference observations or adopt a constrained correction scheme, such as the constrained adaptive bias correction (Han and Bormann, 2016). The GMWR assimilation was implemented using 3DVAR, based on RTTOV-gb and WRFDA, and only static background-error covariances were employed in this study. The background error covariance matrix plays an important role in variational data assimilation, but this type of covariance is climatological, spatially homogeneous, and isotropic. This may limit the impact of GMWR assimilation, and flow-dependent error covariances should be considered in future work.

Moreover, only clear-sky GMWR radiances were assimilated in this study. Since precipitation processes are often accompanied by extensive cloud cover, few clear-sky GMWR observations were available. To better explore the potential of GMWR assimilation, experiments were conducted during periods with abundant clear-sky GMWR data (e.g., a ten-day period in October 2023), which coincided with minimal heavy precipitation. Studies on satellite all-sky assimilation have shown that

incorporating cloud- and precipitation-affected data improves forecasts (Ma et al., 2022; Xian et al., 2019), highlighting the need for future research on all-sky assimilation of GMWRs. Under such conditions, assimilation experiments could be conducted during a different or longer period, given that assimilated GMWR observations would be relatively more abundant. It is noted that GMWRs exhibit higher sensitivity and provide more valuable observations of the lower troposphere and planetary boundary layer compared to satellite-based microwave radiometers (Shi et al., 2023). Building on this study, future research could explore the joint direct assimilation of satellite-based and ground-based microwave radiometers. A more comprehensive evaluation of upper-air impacts could also be performed using additional independent observations, including aircraft reports and radio occultation data.

## Appendix A: Observed versus simulated Tb

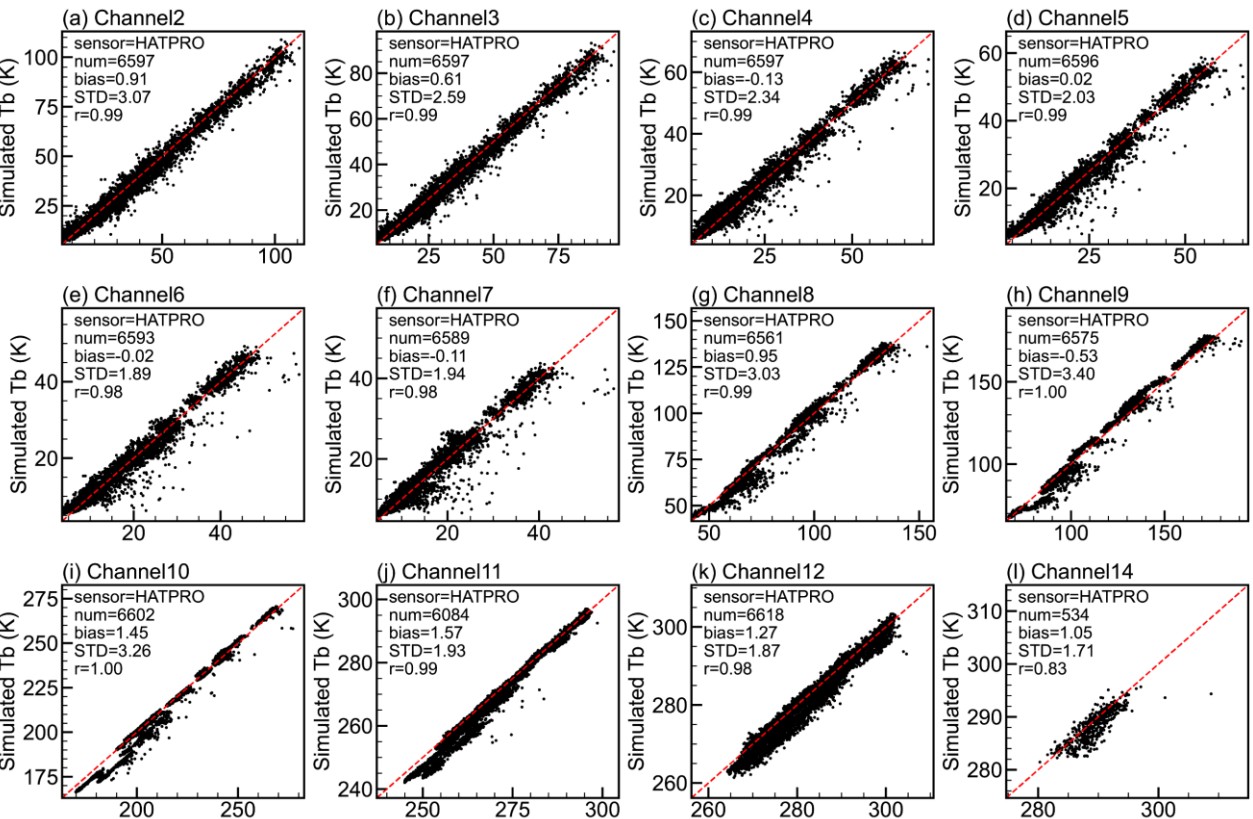

Figure A1. Scatterplots of observed brightness temperature (Tb) versus simulated Tb for HATPRO. Same as Fig. 3 but for additional channels.

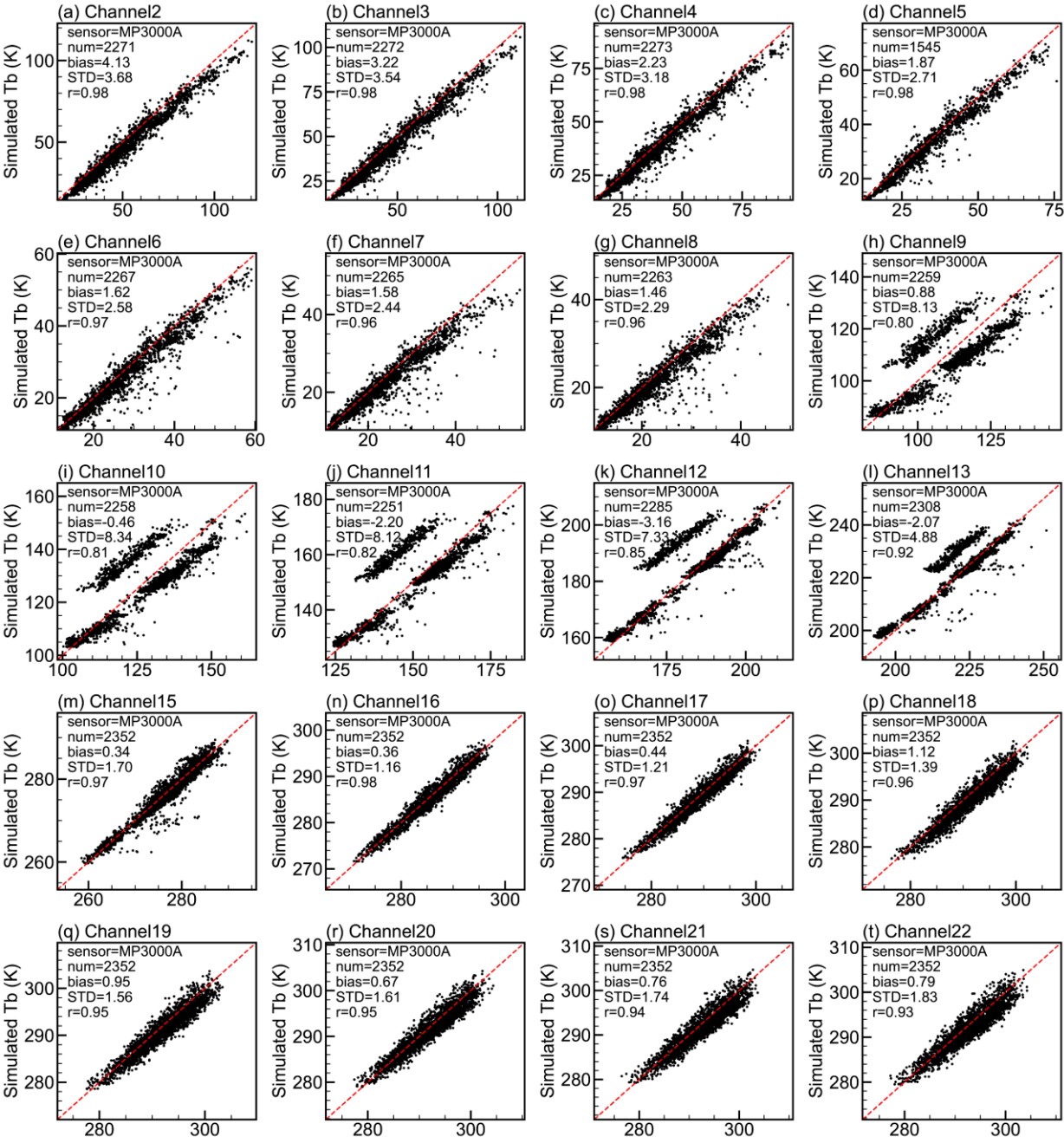

Figure A2. Scatterplots of observed brightness temperature (Tb) versus simulated Tb for MP3000A. Same as Fig. 3 but for additional channels.

**Appendix B: PDF distributions of O−B**

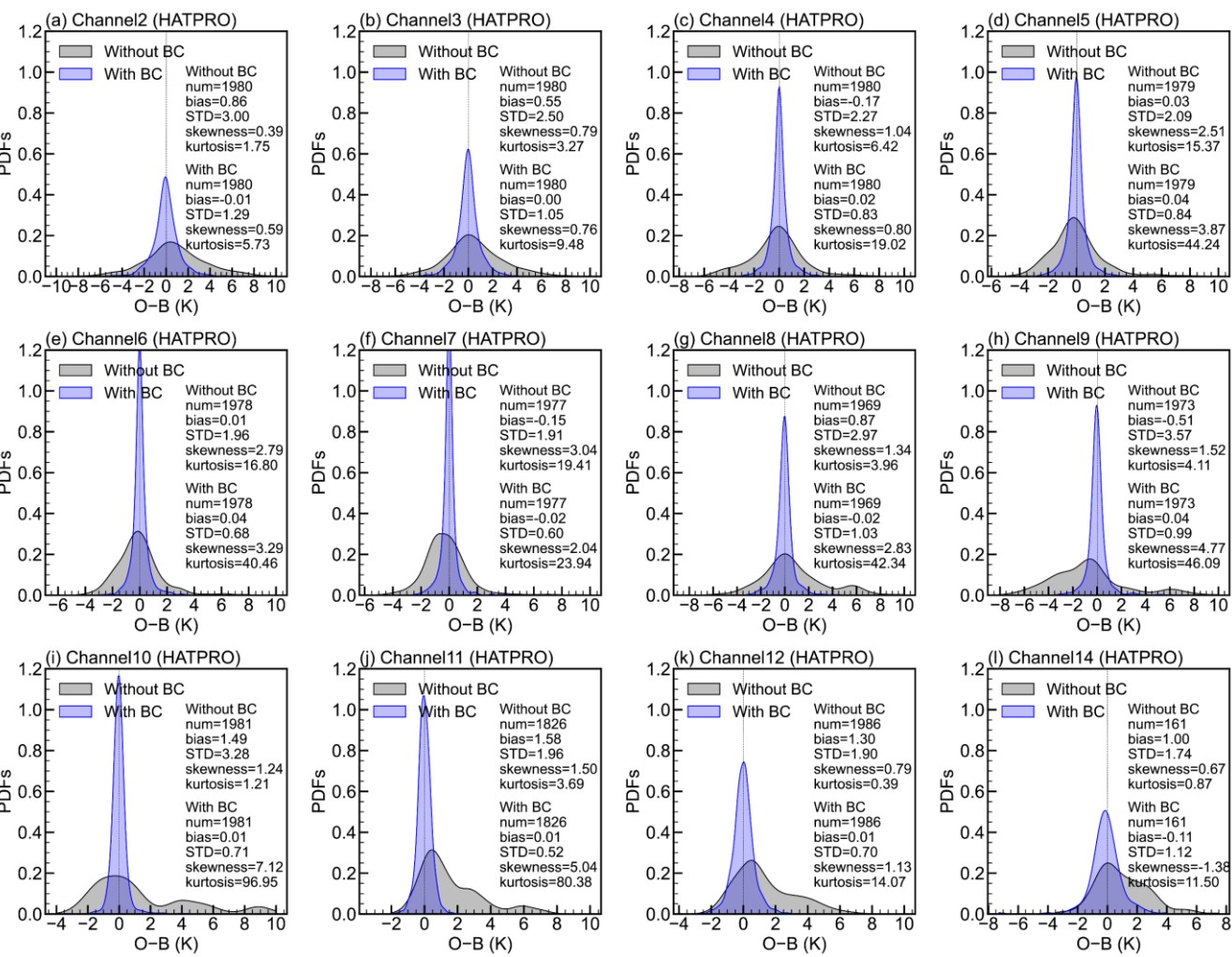

Figure B1.  Probability density functions (PDFs) of the O−B distributions for HATPRO. Same as Fig. 6 but for additional channels.

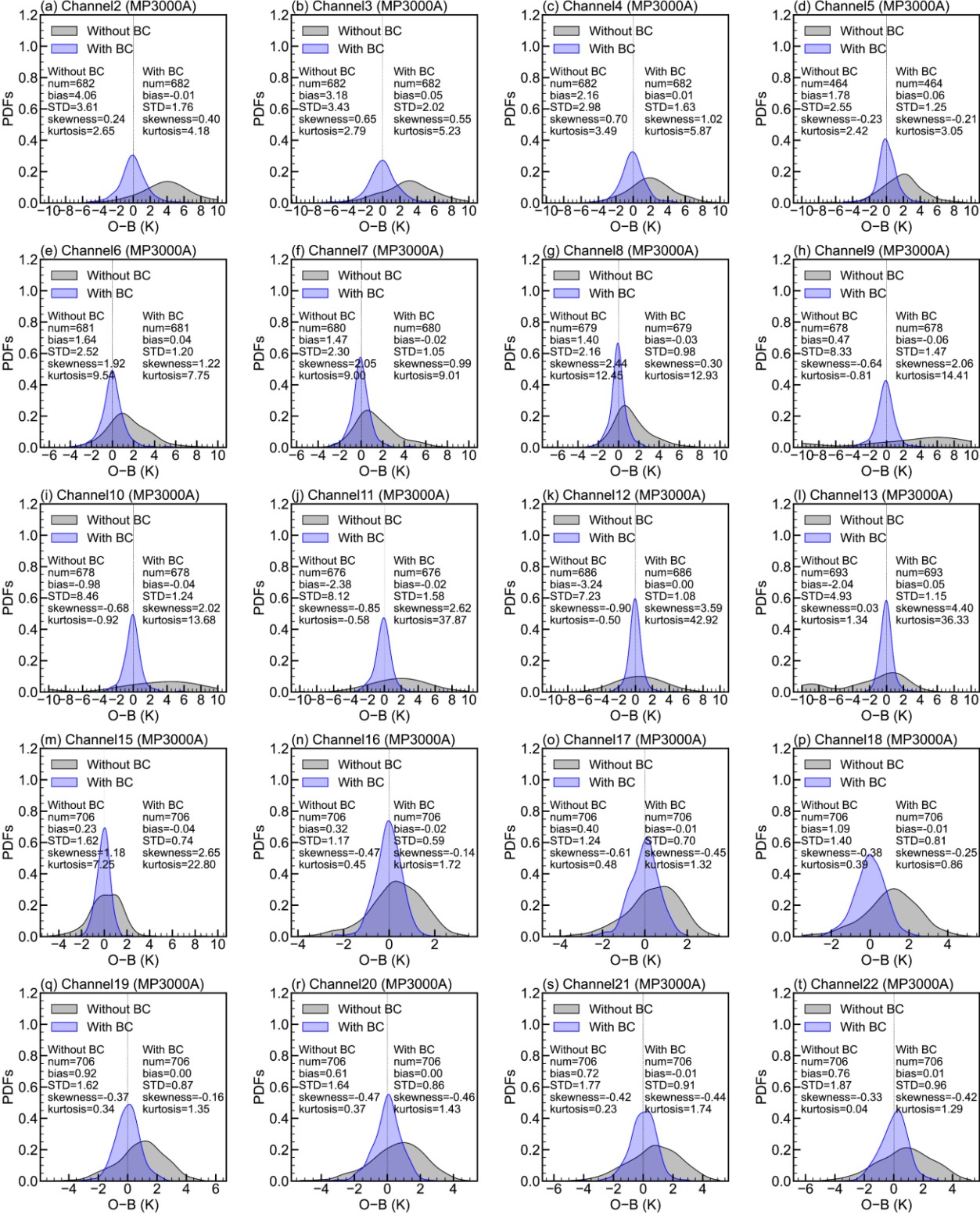

Figure B2. Probability density functions (PDFs) of the O−B distributions for MP3000A. Same as Fig. 6 but for additional channels.

**Code availability**

The RTTOV-gb v1.0, WRF v4.5, WRFDA v4.5, along with the code for developing a direct assimilation module for GMWR radiances and training a machine learning-based GMWR bias correction model, are available on Zenodo (https://doi.org/10.5281/zenodo.16731169; Zheng et al., 2025a)

580 **Data availability**

The GMWR data and precipitation analysis product are provided by the Chinese Meteorological Administration and can be obtained via request from https://www.cma.gov.cn/en/. The AGRI CLM used are available at https://satellite.nsmc.org.cn/portalsite/default.aspx?currentculture=en-US. The NCEP FNL data used are available at https://rda.ucar.edu/datasets/d083003/. The assimilated GTS data are available at https://rda.ucar.edu/datasets/d337000/. The 585 model outputs for the single-observation assimilation experiment and the three-month sample dataset for the "Machine learning based bias correction for GMWR" section is available on Zenodo (https://doi.org/10.5281/zenodo.14586346; Zheng et al., 2025b)

**Author contribution**

WS conceived the idea and designed the research. QZ performed the research and wrote the first draft of the manuscript. All 590 authors discussed the results and contributed to writing and revisions.

**Competing interests**

The authors declare no conflicts of interest or competing financial interests.

**Acknowledgements**

The authors gratefully acknowledge Dr. Domenico Cimini for his help in usage of RTTOV-gb. They also thank the Numerical 595 Weather Prediction Satellite Application Facility (NWP SAF) for providing the source code of RTTOV-gb v1.0, and the National Center for Atmospheric Research (NCAR) for providing the source code of WRF v4.5 and WRFDA v4.5. Additionally, the authors appreciate the National Centers for Environmental Prediction (NCEP) for providing the FNL data,

the China Meteorological Administration (CMA) for supplying the GMWR data and precipitation analysis product, and the National Supercomputing Center in Chengdu for their computational support.

## Financial support

This study was jointly supported by the National Key Research and Development Program of China (2023YFC3007504), State Key Laboratory of Severe Weather Meteorological Science and Technology (2025QZA06), the National Natural Science Foundation of China (42475013 and U2442214), Xizang Science and Technology Major Program (XZ202402ZD0006-04), Sichuan Science and Technology Program (2024YFFK0110), Southwest Sichuan (Ya'an) Rainstorm Laboratory (CXNBYSYSZD202401).

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
