# Peer review of "Direct assimilation of ground-based microwave radiometer observations with machine learning bias correction based on developments of RTTOV-gb v1.0 and WRFDA v4.5"

_EGUsphere, 2025_

## Author Comment (AC2)

Dear Editors and Reviewers:

We deeply appreciate your helpful comments and suggestions, which enabled us to improve the quality of our present study. We have made revisions and replied to all the comments. Please find the point-by-point responses to the comments below. The comments are given below in black, our responses are in blue, and proposed changes to the manuscript are in red.

**Reply to Reviewer #1**:

1. Firstly, I'd like to have more details about the 10-day period over which the assimilation experiments were conducted. Unless I'm mistaken, it's implicitly understood that this is a clear-sky period (especially when the cloud cover mask is mentioned), but this is explicitly stated only at the end of the article. On the other hand, I think it would have been nice to provide more justification for this particular period.

Response:

Thanks for your comments. The assimilation experiment was conducted under clear-sky conditions. Figure R1 illustrates clear-sky observation counts in the target region of Southwest China from the microwave radiometer (MWR). Among the available MWR observations spanning August to October 2023, a ten-day period (highlighted in blue in Figure R1c) from 13 October 2023 to 22 October 2023 exhibited a notably higher frequency of clear-sky MWR observations. As an initial study that mainly focused on direct assimilation of clear-sky MWRs, this period can better evaluate the impact of MWR assimilation. More details about the 10-day period over which the assimilation experiments have been added in the manuscript as below.

"The assimilation experiments were conducted under clear-sky conditions due to the uncertainties in the model and observation operators under cloudy or rainy conditions. All experiments were conducted over a ten-day period from 13 to 22 October 2023. Among the available GMWR observations from August to October 2023, this period exhibited a notably higher frequency of clear-sky data, which was more favorable for demonstrating the role and potential of GMWR

assimilation. Before implementing bias correction, clear-sky screening, first-guess departure check, and whitelist check were sequentially applied to improve measurement quality. Subsequently, a relative departure check was applied prior to minimization. For the 6 h, 3 h, and 1 h assimilation intervals, 34 (0.91%), 70 (1.42%), and 76 (0.72%) observations were rejected, respectively. The detailed procedure prior to a single assimilation cycle is as follows:

(1) Observation Selection: The observation nearest to the analysis time within ±10 minutes is selected.

(2) Clear-sky Screening: Clear-sky GMWR observations were screened using the AGRI-based CLM, with background-simulated cloud liquid water path equal to zero.

(3) First-Guess Departure Check: Observations with (O−B) values greater than 20 K are excluded.

(4) Whitelist Check: Remove observations from stations identified as unreliable or displaying abnormal behavior.

(5) Bias Correction: a machine learning bias correction scheme was applied (see Section 3.2).

(6) Relative Departure Check: Applied when the absolute value of the O−B exceeds three times the standard deviation of the observational error, further rejecting questionable data."

[Figure]

Figure R1. Clear-sky observation counts in the target region of Southwest China from the microwave radiometer (MWR) for (a) August, (b) September, and (c) October 2023. The blue line depicts hourly MWR counts, and the red line represents the 6-hour rolling average. The blue shaded region marks the 10-day period selected for assimilation experiments.

2. Still on the subject of this 10-day period, I'd like to know if you've carried out assimilation experiments over other periods? If so, what were the results equivalent? Why wasn't a longer period considered?

Response:

Due to cloud cover and rainfall, the number of clear-sky observations during other periods is small, except for this ten-day period. This study primarily focuses on implementing RTTOV-gb within WRFDA and conducting clear-sky assimilation based on machine learning bias correction. To this end, we mainly concentrate on assimilation during this specific period. As demonstrated in this paper, clear-sky assimilation of MWRs has significant potential to improve numerical forecasts. It would be interesting to implement cloudy-region assimilation for MWRs in the next step, as it could incorporate more MWR

observations than clear-sky assimilation. Under such conditions, assimilation experiments would be conducted over other periods or extended to a longer period, given that assimilated MWR observations would be relatively more abundant. Nevertheless, a discussion about experiment periods has been added in the discussion as below.

"Moreover, only clear-sky GMWRs were assimilated in this study. Since precipitation processes are often accompanied by extensive cloud cover, few clear-sky GMWRs were available. To better explore the potential of GMWR assimilation, experiments were conducted during periods with abundant clear-sky GMWRs (e.g., a ten-day period in October 2023), which coincided with minimal heavy precipitation. Studies on satellite all-sky assimilation have shown that incorporating cloud- and precipitation-affected data improves forecasts (Ma et al., 2022; Xian et al., 2019), highlighting the need for future research on all-sky assimilation of GMWRs. Under such conditions, assimilation experiments could be conducted during a different or longer period, given that assimilated GMWR observations would be relatively more abundant."

3. Regarding the single observation experiment, I'd like to know why the specific humidity analysis increments aren't totally isotropic (although they're close) as they are for temperature.

Response:

In the single observation experiment, the background error covariance used in WRFDA is CV5. Pseudo relative humidity (RHs) is the control variable in CV5 and is minimized during assimilation. However, RHs is not a model variable, and its analysis increment is transformed into model variables (e.g., water vapor mixing ratio). During this transformation, the water vapor mixing ratio may not remain fully isotropic. As shown in Figure R2, while the analysis increment of RH appears isotropic, this is not the case for the water vapor mixing ratio.

[Figure]

Figure R2 The horizontal analysis increments for (a) water vapor mixing ratio and (b) relative humidity in single-observation assimilation experiment.

4. My final comment concerns Figures 10 and 12. I think it would have been clearer to present the RMSE or FSS values directly, rather than the differences with the control experiment. I understand that this removes a curve from the graphs and perhaps improves readability. But having the RMSE values would be informative about the errors made by the model.

Response:

The RMSE and FSS for the CNTL experiment are shown as black solid lines in Figures 10 and 12. The corresponding changes we have implemented are as follows:

"The time series of RMSE for the CNTL experiment and RMSE differences (assimilation experiments minus the CNTL experiment) against surface station observations for 2 m temperature, 2 m relative humidity, and 10 m wind fields are shown in Fig. 11. In the CNTL experiment, the RMSE of temperature and relative humidity initially decreases and then increases with lead time, while the RMSE of the wind field exhibits the opposite trend, increasing at first and then decreasing. The mean RMSEs over the 24-hour forecast period are 2.32 K for temperature, 16.26% for relative humidity, 1.92 m s⁻¹ for zonal wind, and 2.08 m s⁻¹ for meridional wind. Regarding assimilation impacts, the RMSE

reduction for temperature gradually decreases, approaching zero at a lead time of 6 hours, with higher assimilation frequency (GMWR_1H) achieving a greater RMSE reduction.

[Figure]

Figure 1: Verification of the forecast against surface station observations, based on the ten-day assimilation experiment conducted from 13 to 22 October 2023. RMSE (black line) for the CNTL experiment and RMSE differences (colored lines) between the assimilation experiments and the CNTL experiment for (a) temperature, (b) relative humidity, (c) zonal wind, and (d) meridional wind.

Figure 13 presents the time series of FSS for the CNTL experiment and FSS differences (assimilation experiments minus the CNTL experiment). The assimilation experiments were conducted during a period characterized by a higher frequency of clear-sky observations. Cloud cover and precipitation were limited throughout the 10-day period, resulting in the absence of frequent heavy rainfall events. Consequently, the FSS was calculated using small precipitation thresholds. In the CNTL experiment, the FSS for 3 h accumulated precipitation shows an initial decline followed by a subsequent increase with lead time, with relatively low FSS values observed around the 9 h forecast period. Moreover, the FSS generally decreases as the precipitation threshold increases. The time

mean FSS values are 0.47, 0.45, 0.42, and 0.39 for thresholds of 3 mm, 4mm, 5mm, and 6 mm, respectively. Regarding the role of GMWR assimilation in precipitation forecasting, the results indicate that assimilating GMWR radiances enhances precipitation forecasts, with FSS differences increasing progressively at higher precipitation thresholds.

[Figure]

Figure 13: The time series of FSS (black line) for CNTL experiment and FSS differences (colored lines) between the assimilation experiments and the CNTL experiment. These experiments were conducted from 13 to 22 October 2023.The FSS was calculated for 3 h accumulated precipitation for thresholds of (a) 3 mm, (b) 4 mm, (c) 5 mm, and (d) 6 mm.

---

## Author Comment (AC3)

Dear Editors and Reviewers:

We deeply appreciate your helpful comments and suggestions, which enabled us to improve the quality of our present study. We have made revisions and replied to all the comments. Please find the point-by-point responses to the comments below. The comments are given below in black, our responses are in blue, and proposed changes to the manuscript are in red.

**Reply to Reviewer #2**:

General Comments

1. Could you please include a few more details about the assimilation framework. Specifically:

**-Timing of Observations:** Do you select the observation that most closely matches the analysis time, or do you integrate over a broader time window?

**-Data Processing Before Bias Correction:** Aside from using the cloud mask, do you apply any additional procedures to improve measurement quality before performing bias correction?

**-QC**: Do you use any quality-control checks (e.g., discarding observations based on O−B thresholds), and if so, how many observations are rejected?

Response:

The observation nearest to the analysis time within ±10 minutes will be selected.

Before implementing bias correction, we sequentially applied cloud and precipitation detection, O−B (observation-minus-background) checks, and quality control using a whitelist. Cloud and precipitation detection were based on the FY-4B cloud mask and rain gauge data from MWRs. The O−B check served as a preliminary screening step that excluded observations with absolute O−B values greater than 15 K. The whitelist was established to identify reliable station observations based on the correlation between simulated and observed brightness temperatures (BTs). Specifically, we evaluated the correlation between observed and simulated BTs at each station. Stations where observed BTs did not vary consistently with simulated BTs—indicated by low correlation

coefficients—were classified as "problematic stations." Some of these problematic stations returned to normal after specific dates, and observations from those periods were retained.

After bias correction, additional quality control based on O−B was conducted, including a three-sigma observational error check. For the six-hour, three-hour, and one-hour assimilation intervals, the total numbers of observations were 3,724, 4,946, and 10,619, respectively. Following a three-sigma observational error check, 34 (0.91%), 70 (1.42%), and 76 (0.72%) observations were rejected. The corresponding changes we have implemented are as follows:

"Before implementing bias correction, clear-sky screening, first-guess departure check, and whitelist check were sequentially applied to improve measurement quality. Subsequently, a relative departure check was applied prior to minimization. For the 6 h, 3 h, and 1 h assimilation intervals, 34 (0.91%), 70 (1.42%), and 76 (0.72%) observations were rejected, respectively. The detailed procedure prior to a single assimilation cycle is as follows:

(1) Observation Selection: The observation nearest to the analysis time within ±10 minutes is selected.

(2) Clear-sky Screening: Clear-sky MWR observations were screened using the AGRI-based CLM, with background-simulated cloud liquid water path is zero.

(3) First-Guess Departure Check: Observations with (O−B) values greater than 20 K are excluded.

(4) Whitelist Check: Remove observations from stations identified as unreliable or displaying abnormal behavior.

(5) Bias Correction: a machine learning bias correction scheme was applied (see Section 3.2).

(6) Relative Departure Check: Applied when the absolute value of the O−B exceeds three times the standard deviation of the observational error, further rejecting questionable data."

2. In Chapter 3, the O−B statistics are examined in detail. From the scatter plots,

it appears that the K-band O−B distribution may be bimodal for both radiometer types—a potential issue for 3D-Var, which typically assumes unimodal (Gaussian) errors. After showing the initial scatter plots of BT(sim) vs. BT(obs), the paper primarily focuses on bias and standard deviation. It would be very interesting to see if the bias correction addresses this bimodality. I encourage you to present histograms or PDFs of the O−B errors before and after the bias correction is applied. I would also encourage you to comment briefly on the skew and kurtosis of the distributions.

Response:

The bimodality in the O−B distribution has been addressed to some extent through bias correction. The probability density functions (PDFs) of the O−B have been included in the manuscript, along with the skewness and kurtosis of the distributions. The corresponding changes we have implemented are as follows:

"From the scatter plots in Fig. 3, the O−B distribution appears bimodal—an issue that may affect 3D-Var, which typically assumes the errors to be unimodal (Gaussian). Similar to Fig. 3, channel 1 (K-band) and channel 13 (V-band) of HATPRO, as well as channel 1 (K-band) and channel 14 (V-band) of MP3000A, are selected for detailed analysis (Fig. 6). Results for the remaining channels are presented in Figures B1 and B2. The biases for HATPRO channel 1, HATPRO channel 13, MP3000A channel 1, and MP3000A channel 14 are 1.24 K, 2.21 K, 3.00 K, and –0.64 K, respectively, with corresponding STDs of 3.38 K, 2.90 K, 3.89 K, and 3.08 K. The differences between the test set and the full dataset (shown in Fig. 3) are negligible, with a maximum bias difference of 0.10 K and a maximum STD difference of 0.08 K, highlighting the strong representativeness of the test set. From the PDF distributions of O–B, both instruments exhibit a positive bias in the K-band with a unimodal distribution. In contrast, a bimodal distribution is observed in the V-band: the second peak appears on the right for HATPRO and on the left for MP3000A. These results are consistent with the scattering patterns shown in Fig. 3. After the bias correction is applied, both the bias and STD are reduced, and the O–B

distribution becomes more sharply concentrated around zero, accompanied by an increase in kurtosis. For example, in channel 1 of MP3000A, the bias and STD decrease from 1.24 K and 3.38 K to 0.03 K and 1.44 K, respectively, while the kurtosis increases markedly from 1.53 to 9.44. It is also noteworthy that the bimodal distributions in the V-band for both instruments become unimodal after the correction. Meanwhile, the skewness decreases from 1.04 and 1.54 to 0.55 and 1.05, respectively, indicating a more symmetrical O–B distribution. These results demonstrate that the proposed bias correction scheme effectively reduces bias and STD, addresses bimodal distribution, and shifts the O–B distribution closer to a Gaussian shape.

[Figure]

Figure 6: Probability density functions (PDFs) of the O−B distributions, based on a test set randomly selected from 30% of the three-month sample dataset collected from August to October 2023. The top and bottom rows correspond to the HATPRO and MP3000A sensors, respectively, while the left and right columns represent the K-band and V-band. Each panel displays the number of samples (num), the mean (bias),

standard deviation (STD), skewness, and kurtosis of the distributions."

3. The FSS improvements of rain rate seem impressive to say that the only change here is ground based radiometers. I suppose that the reason any impact is visible is only after 9 hours is because cloudy data is excluded. It may be interesting to show some statistics of the rainfall events in the 10-day period. Is one heavy precipitation event responsible for this improvement in forecast skill? Do you intend to repeat the experiment including cloudy data?

Response:

Thank you for your valuable suggestion. Upon re-checking the FSS calculation, we found that the improvement reported in the manuscript may not be as substantial as initially perceived.

The assimilation experiment period was selected to coincide with more frequent clear-sky conditions to maximize the availability of GMWR observations. During this period, precipitation was minimal, with no particularly intense precipitation events. Furthermore, during the phase of substantial FSS improvement (after 9 hours), observed precipitation was nearly absent. Although simulated precipitation exceeded observed amounts, it remained low overall, with false alarms being the primary source of error. The limited amounts of both observed and simulated precipitation led to a small sample size in the FSS calculation, resulting in the division of two small values. This mathematical characteristic amplified fluctuations in FSS scores, producing an apparent and pronounced improvement.

To mitigate the effect of small sample sizes in the FSS calculation described above, the thresholds for calculating FSS scores were lowered from 6–15 mm to 3–6 mm. The GMWR assimilation in this study, which is limited to clear-sky areas, does not utilize information from cloudy regions, resulting in reduced data usage—particularly during precipitation events. In future work, assimilation techniques for cloudy areas will be developed and focus on improving forecasts of heavy precipitation processes.

We revised the thresholds for FSS and provided clarification. The

corresponding text has been revised as follows:

" The assimilation experiments were conducted during a period characterized by a higher frequency of clear-sky observations. Cloud cover and precipitation were limited throughout the 10-day period, resulting in the absence of frequent heavy rainfall events. Consequently, the FSS was calculated using small precipitation thresholds. In the CNTL experiment, the FSS for 3 h accumulated precipitation shows an initial decline followed by a subsequent increase with lead time, with relatively low FSS values observed around the 9 h forecast period. Moreover, the FSS generally decreases as the precipitation threshold increases. The time mean FSS values are 0.47, 0.45, 0.42, and 0.39 for thresholds of 3 mm, 4mm, 5mm, and 6 mm, respectively. Regarding the role of GMWR assimilation in precipitation forecasting, the results indicate that assimilating GMWR radiances enhances precipitation forecasts, with FSS differences increasing progressively at higher precipitation thresholds. Additionally, increasing assimilation frequency shows the potential to further enhance forecast performance. When assimilating GMWR data at a 1 h frequency, the time-averaged FSS improvements for 3 h accumulated precipitation are 0.02 (3.9 %) for the 3 mm threshold, 0.02 (4.7 %) for 4 mm threshold, 0.03 (7.3 %) for 5 mm threshold, and 0.04 (10.2 %) for 6 mm threshold precipitation. For 3 h accumulated precipitation with a threshold of 6 mm, the time-averaged FSS improvements are 0.01, 0.02, and 0.03 for GMWR_6H, GMWR_3H, and GMWR_1H, respectively. These findings are consistent with the above verification against radiosonde and surface station observations, suggesting that GMWR assimilation can improve forecasts and that higher-frequency assimilation leads to further enhancements.

[Figure]

Figure 13: The time series of FSS (black line) for CNTL experiment and FSS differences (colored lines) between the assimilation experiments and the CNTL experiment. These experiments were conducted from 13 to 22 October 2023.The FSS was calculated for 3 h accumulated precipitation for thresholds of (a) 3 mm, (b) 4 mm, (c) 5 mm, and (d) 6 mm."

4.   Could you add some information about how the background error covariance matrix was generated? Was any testing done to optimise this? Similarly, it would be helpful to know how you specified the observation-error covariances—for instance, the assumed observation-error standard deviations or correlations.

Response:

The background error covariance matrix was generated using the Generalized Background Error Covariance Matrix Model (GEN_BE v2.0) with the National Meteorological Center (NMC) method. Specifically, CV5, which is commonly used in WRFDA, was employed in this study, and the length scale was tuned using a single-observation experiment. Observation-error correlations are typically assumed to be zero in WRFDA, resulting in a diagonal observation-error covariance matrix. Observation errors were specified based on the standard deviation of O–B. More information about background error and observation error have been added in the manuscript as below.

"The static background error covariance for the variational experiments is estimated using the National Meteorological Center (NMC) method (Parrish and Derber, 1992), which uses the difference between WRF forecasts at lead times of 24 h and 12 h (T + 24 h minus T + 12 h) valid at the same time over a specified period. Control variables option 5 (CV5) is adopted for the background error covariance used in 3DVAR. CV5 is domain-dependent and therefore must be generated based on forecast or ensemble data over the same domain. It utilizes streamfunction, unbalanced velocity potential, unbalanced temperature, unbalanced surface pressure, and pseudo relative humidity. In this study, the background error covariance matrix was generated using the Generalized Background Error Covariance Matrix Model (GEN_BE v2.0) (Descombes et al., 2015) based on one month of WRF forecasts. Observation-error correlations are typically assumed to be zero in WRFDA, resulting in a diagonal observation-error covariance matrix. Observation errors were specified based on the standard deviation of O–B."

Specific Comments

5. Figure 1: Was the south-west China domain pre-defined in advance of the study? Were these conditions defined from the border of the computation domain or otherwise? Please justify.

Response:

"Similar to previous studies (Jiang et al., 2017; Nie and Sun, 2023), the target region of Southwest China in this study is defined as the area within the rectangular domain 22°–35°N, 93°–110°E (Fig. 1). This region encompasses the Hengduan Mountains, the Yunnan–Guizhou Plateau, and the Sichuan Basin, and is generally consistent with Chinese administrative divisions."

6. line 106: "However, MWR radiances are upward-looking microwave observations, which differ from the downward-looking observation of satellites." I find this sentence quite jarring. Before in the article you refer to "ground-based MWRs", so when you state that "MWR radiances are upward-looking" it seems

like you refer to the microwave radiometer instrument in general, not simply ground based microwave radiometers. I would change the sentence to refer simply to the platform (ground-based vs satellite) and not contrast microwave radiometer with satellite as this doesn't make sense.

Response:

To avoid confusion between ground-based microwave radiometers and microwave radiometers (MWRs) in general, the term "ground-based microwave radiometers (GMWRs)" is now used throughout this study. Accordingly, the sentence has been revised:

"RTTOV, a fast RTM, is widely used for assimilating satellite radiance data. However, GMWR radiances are upward-looking microwave observations, differing from the downward-looking measurements of satellite-borne microwave radiometers."

7. Line 134: Could you say (at some point in the paper, not necessarily here) when the three month training data for your bias correction algorithm was taken?

Response:

"To estimate the bias and develop a bias correction scheme for GMWR direct assimilation, a long-term experiment was conducted from August to October 2023, yielding a three-month sample dataset."

"The flowchart illustrating the training and evaluation process of the bias correction (BC) model is shown in Fig. 5b. The three-month sample dataset (described in Section 3.1) was randomly split into a training set (70 %) and a test set (30 %)."

8. Figure 2: Could you explain the plot axis label d Tans/ d ln P, I am not familiar with weighting functions of this type. Are the temperature/humidity increments representative of all data assimilation experiments? If not, please elaborate in the text. For plot c,d,e and f please label the colourbars and make the magnitude (1eX) more evident.

Response:

Transmittance $\tau$ varies monotonically with pressure $p$. The downwelling emission term can be expressed as:

$$L = \int_{\tau_t}^{1} B\,(T)\,d\tau = \int_{p_t}^{p_s} B\,(T)\,\frac{\partial \tau}{\partial lnp}\,dlnp$$

where $L$ is the radiance at the ground, $B$ is the Planck function, $T$ is temperature, $\tau_t$ is the transmittance from the surface to the top of the atmosphere, $p_t$ is the pressure at the top of the atmosphere, $p_s$ is the surface pressure.

The weighting function, $\frac{\partial \tau}{\partial lnp}$, which is the derivative of transmittance with respect to height (pressure), describes the relative contribution of each atmospheric layer to the total radiation emitted to the surface (Cui et al. 2020; Thépaut, 2023). In figure 2, the axis label "d Tans/ d ln P" denote derivative of transmittance with respect to vertical coordinate, d(transmittance) / d(log(p)). The temperature and humidity increments are obtained from single-observation assimilation experiments, which are conducted to confirm that the GMWR direct assimilation module performs correctly. Although this experiment is not representative of all data assimilation experiments, they provide valuable insights into the characteristics of GMWR assimilation. Figure 2 and the corresponding text have been revised to improve clarity.

"According to Cui et al. (2020), WFs are calculated as the derivative of transmittance with respect to the natural logarithm of pressure.

It should also be noted that this experiment was conducted to verify the correct performance of the GMWR direct assimilation module and to provide valuable insights into the characteristics of GMWR assimilation. However, it is not representative of the subsequent multi-observation, multi-channel assimilation experiments.

[Figure]

Figure 2: Normalized weighting functions of (a) HATPRO and (b) MP3000A calculated using the RTTOV-gb. The (c, d) horizontal and (e, f) vertical analysis increments for (c, e) temperature and (d, f) water vapor mixing ratio in single-observation assimilation experiment. The vertical increments are cross-sections along the green lines shown in the horizontal increments. The colorbar tick labels for temperature and water vapor mixing ratio are expressed in scientific notation as $1\times10^{-2}$ and $1\times10^{-4}$, respectively."

Thépaut, J. N., 2003: Satellite data assimilation in numerical weather prediction: An overview. *Proc. ECMWF Seminar on Recent Developments in Data Assimilation for Atmosphere and Ocean*, Reading, UK, ECMWF, 75–96.

Cui, X., Z. Yao, Z. Zhao, Y. Zhai, Z. Sun, W. Cheng, and C. Gu, 2020: Use of Double Channel Differences for Reducing the Surface Emissivity Dependence of Microwave Atmospheric Temperature and Humidity Retrievals. *Earth and Space Science*, **7**, e2019EA000854, https://doi.org/10.1029/2019EA000854.

9. Line 160: "For HATPRO, more than 6,000 samples are analyzed. The O−B biases are 1.25 K for the K band (channel 1) and 2.14 K for the V band (channel 13)". Are these statistics the average for the whole band or that particular channel? If they are for the whole band, make this explicit on Figure 3. If not, then these results should not be generalised to the whole band.

Response:

These statistics are for particular channel and the corresponding text have been revised.

10. Figure 3: Why were these particular channels selected. Could you comment in the text about whether these plots are representative of the whole band? It would also be nice if you could include the same plots for the other channels in the appendix.

Response:

These plots are not representative of the whole band. The same plots for the other channels are added in the appendix. And the corresponding text have been revised as following:

"A comparative scatterplot analysis of observed and simulated brightness temperatures was conducted. For most channels, the scatter points are closely aligned along the diagonal and exhibit high correlation coefficients, indicating strong agreement between the simulations and observations. However, the scatter for some channels forms two distinct clusters. To further investigate, representative channels from the K-band (water vapor absorption lines) and the V-band (temperature-sensitive oxygen absorption lines) were selected. Figure 3 presents scatterplots for channel 1 (K-band) and channel 13 (V-band) of HATPRO, and channel 1 (K-band) and channel 14 (V-band) of MP3000A. Results for the remaining channels are shown in Figures A1 and A2. For HATPRO, more than 6,000 samples are analyzed for channels 1 and 13. The O−B biases are 1.25 K for channel 1 and 2.14 K for channel 13, with standard deviations (STD) of 3.35 K and 2.82 K, respectively. Additionally, the scatter distribution for channel 13 is not centered, showing a cluster shifted to the right of the diagonal (Fig. 3b). For MP3000A, more than 2,000 samples are analyzed for channels 1 and 14, with O−B biases of 3.06 K for channel 1 and −0.54 K for channel 14. The O−B STDs are 3.94 K and 3.08 K, respectively. Similar to the results for HATPRO channel 13 (V-band), the scatter for MP3000A channel 14 (V-band) also shows a cluster offset from the diagonal, but to the left (Fig. 3d).

Based on these results, significant O−B biases are detected in GMWR observations, with their characteristics varying across different sensors and channels. However, the correlation coefficients between observed and simulated brightness temperatures are high, at least 0.95, suggesting that these biases can be effectively corrected.

[Figure]

Figure A1: Scatter plot of observed brightness temperature (Tb) versus simulated Tb for HATPRO. Same as Fig. 3 but for additional channels.

[Figure]

Figure A2: Scatter plot of observed brightness temperature (Tb) versus simulated Tb for MP3000A. Same as Fig. 3 but for additional channels."

11. With respect to the apparent two clusters in the V band, were the (clustered) offset values consistently from the same radiometers. Do the offsets also correspond to a particular time period (e.g. before vs after calibration) or weather condition (after rain, in direct sunlight)?

Response:

The apparent two clusters in the V band correspond to a cluster shifted to the right of the diagonal for channel 13 in HATPRO, and a cluster shifted to the left of the diagonal for channel 14 in MP3000A. These offsets correspond to

positive and negative O–B deviations, respectively. According to Figure 4, the positive O–B values originate from GMWRs at stations 56312, 56029, and 55664, while the negative O–B values are primarily from station 57461. The offset was consistently present and did not correspond to a particular time period.

12. line 182: "Regarding the correlation coefficients between observed and simulated brightness temperatures, the overall values are high but slightly lower for channels 4 to 9." What is high? Please be more precise.

Response:

This sentence was revised as following.

"The correlation coefficients between observed and simulated brightness temperatures are high across all channels (typically above 0.90), although they are slightly lower for channels 4 to 9."

13. Figure 4: Please add colourbar axis label and units for all sub plots. For HATPRO channel 14, values are mainly white, but this colour is not included in the colourbar- does that mean that the values are out of range, missing or otherwise?

Response:

The colorbar axis label and units have been added. At some stations, HATPRO did not observe channel 14, resulting in missing data that originally appeared as white areas in the figure and were not represented in the colorbar. These areas have been changed to grey, with an explanatory note added accordingly.

"

[Figure]

Figure 4: Statistics at each station based on samples collected from August to October 2023. O−B (a) bias and (b) standard deviations (STD) for HATPRO; (c) correlation coefficient (r) between observed and simulated brightness temperatures for HATPRO; (d–f) same as (a–c) but for MP3000A. Some stations did not provide observations for specific channels; the corresponding missing data are displayed in grey in the figure."

14. Line 215: "These hyperparameters were tuned using GridSearchCV"
Please properly reference scikit learn (or the library in question) when refering to this.

Response:

The reference for scikit-learn has been added.

15. Line 235: ". Furthermore, some individual channels, such as channel 10, display bimodal distributions."

As stated above, it would be interesting to see these plots.

Response:

These plots have been added in appendix.

"For most channels, the PDFs exhibit a unimodal pattern, with peak positions deviating from zero, indicating that the O−B values are biased. For some channels, the distributions are multimodal, characterized by a secondary peak superimposed on the primary one. Although these issues are present in the original distributions, after bias correction, the PDFs approximate an unbiased distribution, and the secondary peaks are effectively suppressed, demonstrating the effectiveness of the correction. From the scatter plots in Fig. 3, the O−B distribution appears bimodal—an issue that may affect 3D-Var, which typically assumes the errors to be unimodal (Gaussian). Similar to Fig. 3, channel 1 (K-band) and channel 13 (V-band) of HATPRO, as well as channel 1 (K-band) and channel 14 (V-band) of MP3000A, are selected for detailed analysis (Fig. 6). Results for the remaining channels are presented in Figures B1 and B2.

[Figure]

Figure B1: Probability density functions (PDFs) of the O−B distributions for

HATPRO. Same as Fig. 6 but for additional channels.

[Figure]

Figure B2: Probability density functions (PDFs) of the O−B distributions for MP3000A. Same as Fig. 6 but for additional channels."

16. Figure 6: temperatrure and water vapour bands are labelled the wrong way around. I'm not sure if a continuous line plot is the most appropriate for the unlinked variables (bottom plot).

Response:

Corrected

17. Line 415: "The RTTOV-gb coefficient files are trained on global profiles and are not tailored to the plateau region; consequently, their vertical coordinates extend up to 1050 hPa, while surface pressure in the plateau region typically exceeds 700 hPa."

Could you please clarify why having the RTTOV-gb coefficient files only extend down to 1050 hPa creates an issue at high-altitude stations? Conceptually, if the sensor is located in a region where the surface pressure is around 700 hPa, that site is "above" the portion of the atmosphere from 1050–700 hPa (which would presumably be below ground in the plateau setting). How does this mismatch in the vertical extent of the RTTOV-gb coefficient files lead to biases for sensors effectively well above 1050 hPa? In other words, why does the "upper pressure limit" become problematic when the actual surface is located at a pressure lower (i.e., a higher altitude) than the coefficient file's assumed maximum pressure?

Response:

RTTOV-gb is trained on global atmospheric profile datasets, where profiles are interpolated onto 101 pressure levels to derive regression coefficients between optical depths and predictors, thereby producing the coefficient files. These global profiles extend to high surface pressures, up to approximately 1050 hPa, enabling the vertical coordinate of the coefficient files to reach this level. However, in plateau regions, surface pressures typically fall below 700 hPa, indicating a substantial climatic difference between the training profiles and those required for simulations in high-altitude areas. This discrepancy may introduce simulation biases when RTTOV-gb is applied over plateaus. Indeed, in satellite data assimilation, efforts have already been made to enhancing fast radiative transfer model using local training profiles (Di et al. 2018). To clarify,

our intention is not to highlight a mismatch in vertical pressure coordinates, but to suggest that the applicability of globally trained coefficients in plateau regions may be limited due to the pronounced climatic divergence. This sentence has been revised accordingly to avoid misunderstanding.

"The RTTOV-gb coefficients are based on global atmospheric profiles, which may differ significantly from the climatic conditions of plateau regions, potentially affecting simulation accuracy."

18. Line 435: "It is noted that satellite-based microwave radiometers are primarily sensitive to the middle and upper atmosphere..."

This isn't technically correct. In atmospheric science, the middle atmosphere generally refers to the stratosphere + mesosphere, i.e. from the tropopause (roughly 8–17 km, depending on latitude) up to ~80–85 km. The upper atmosphere is often taken to be the thermosphere and above. Many operational satellite microwave sensors (e.g., AMSU, MHS, ATMS) retrieve temperature and humidity by sounding channels peaked in the troposphere and lower stratosphere, though they can extend somewhat upward. Their highest sensitivity is thus often in the mid- to upper troposphere, not solely above it.

Response:

This sentence has been revised as following:

"It is noted that GMWRs exhibit higher sensitivity and provide more valuable observations of the lower troposphere and planetary boundary layer compared to satellite-based microwave radiometers (Shi et al., 2023)."

---

## Author Comment (AC4)

Dear Editors and Reviewers:

We deeply appreciate your helpful comments and suggestions, which enabled us to improve the quality of our present study. We have made revisions and replied to all the comments. Please find the point-by-point responses to the comments below. The comments are given below in black, our responses are in blue, and proposed changes to the manuscript are in red.

**Reply to Reviewer #3**:

1. Fig. 8:(1) Suggest adding the 95% significance area of the differences to the figure. Similarly, for all other relevant figures,

Response:

Thank you for your suggestion. To the best of the authors' knowledge, the t-test is commonly used to assess the significance of differences between two independent samples. In previous studies, the difference in mean RMSE was evaluated for statistical significance using a t-test, based on two independent sample sets, each comprising a large number of RMSE values (Lei and Anderson, 2014a; Privé et al., 2014).

As an initial study primarily focused on the direct variational assimilation of GMWR data, the assimilation was limited to clear-sky conditions. All experiments were conducted over a ten-day period from 13 to 22 October 2023. Among the available GMWR observations from August to October 2023, this period exhibited a notably higher frequency of clear-sky data, which was more favorable for demonstrating the role and potential of GMWR assimilation.

In Privé et al. (2014), the RMSE difference profile was calculated by averaging RMSE values from different analysis times over a two-month OSSE experiment. As a result, a large number of RMSE values were available, allowing for the application of a t-test. However, in this study, RMSE was computed over all analysis times as a single aggregated value, rather than being calculated at each analysis time. Therefore, a t-test could not be performed due to the lack of multiple independent samples. On the other hand, due to the limited number of experiments, even if RMSE were calculated at each analysis time, only 10

samples would be available, which could render the test results volatile and statistically unreliable. The all-sky assimilation technique will be the focus of our upcoming study, in which GMWRs from cloudy regions will also be assimilated. At that stage, three-month experiments will be conducted, which will provide a sufficiently large number of samples to enable statistical significance testing of RMSE differences.

Lei, L., and J. L. Anderson, 2014a: Empirical Localization of Observations for Serial Ensemble Kalman Filter Data Assimilation in an Atmospheric General Circulation Model, https://doi.org/10.1175/MWR-D-13-00288.1.

——, and ——, 2014b: Impacts of Frequent Assimilation of Surface Pressure Observations on Atmospheric Analyses, https://doi.org/10.1175/MWR-D-14-00097.1.

Privé, N. C., R. M. Errico, and K.-S. Tai, 2014: The Impact of Increased Frequency of Rawinsonde Observations on Forecast Skill Investigated with an Observing System Simulation Experiment, https://doi.org/10.1175/MWR-D-13-00237.1.

2. (2) It is clear that the MWR data assimilation mostly brings negative impacts to the wind fields. The manuscript needs to cover this fact. The reason for the degradation, as mentioned in the manuscript, may be that the background error covariance does not spread the observation information to the wind field well. Using ensemble covariance may mitigate this.

Response:

In the verification of the initial conditions against radiosonde observations, the assimilation of GMWR data generally exhibits a negative impact on the wind fields. In 3DVAR, the adjoint of RTTOV-gb only updates temperature and humidity in observation space, while adjustments to the wind fields rely on the background error covariance. This suggests that the background error covariance may be a contributing factor to the negative impact on wind field analysis. Therefore, improving the background error covariance (e.g., by incorporating

ensemble-based covariances) could potentially mitigate this limitation. As an initial study primarily focused on the direct variational assimilation of GMWR data with machine learning-based bias correction, it is admitted that this study has some limitations. The MWR assimilation was implemented only using 3DVAR, based on RTTOV-gb and WRFDA. Future work could explore the implementation of MWR assimilation using EnVar, EnKF, or 4DVAR.

The negative impacts to the wind fields has been cover in the results and discussion section of the manuscript, as shown below.

"It is noted that the GMWR assimilation has negative impacts on the wind fields. The RMSE for zonal and meridional winds exhibits a slight negative effect when GMWR is assimilated, with meridional winds even showing an increase in RMSE."

"However, it should be noted that the assimilation of GMWR data generally has a negative impact on the wind fields in the initial conditions. The background error covariance may contribute to this negative impact, as it determines the response of the wind fields to the adjustments in temperature and humidity made by RTTOV-gb. As an initial study primarily focused on the direct variational assimilation of GMWR data with machine learning-based bias correction, it is admitted that this study has some limitations. The GMWR assimilation was implemented using 3DVAR, based on RTTOV-gb and WRFDA, and only static background-error covariances were employed in this study. The background error covariance matrix plays an important role in variational data assimilation, but this type of covariance is climatological, spatially homogeneous, and isotropic. This may limit the impact of GMWR assimilation, and flow-dependent error covariances should be considered in future work."

3. Figures: Suggest adding the time period, when each figures are generated, to the figure captions.
Response:
    Added.

4. Figure 10: This figure shows verifications against surface station observations, right? Suggest changing "station observations" to "surface stations observations", changing subplot titles to "2m Temperature", etc.

Response:

    This figure and the corresponding text have been revised

5. Line 328-329: "the nagavife RMSE differences gradually increase" -> "the RMSE reduction gradually decreases"

Response:

    Corrected

6. Line 330: The RMS difference "decreases" instead of "increases". Consider revising accordingly for all relevant parts.

Response:

    The corresponding text have been revised

7. Fig. 10b: The manuscript is expected to cover the fact that the verification against the surface RH field shows a negative impact from the MWR data assimilation after 12h and explain why.

Response:

    In the verification against surface station observations, the improvement in RH due to GMWR assimilation gradually decreases with increasing lead time and becomes negative after 12 hours. Several factors may be responsible. On one hand, in the verification of initial conditions against radiosonde observations, the assimilation of GMWR data generally exerts a negative impact on the wind fields. The degradation in wind fields may, through model processes, affect the relative humidity. In addition, the model is highly nonlinear, and its errors do not evolve linearly.

    However, the evolution of model errors is complex and difficult to investigate. As this study primarily focuses on the implementation of variational direct assimilation of GMWR data with nonlinear bias correction and provides a

preliminary assessment of its potential impact, the cause of the negative impact on relative humidity after 12 hours will be addressed in detail in future study.

Thanks to the reviewer's suggestion, we have covered this fact in the manuscript as following.

"Similar results are observed for relative humidity, where the RMSE reduction also decreases and approaches zero at a lead time of 12 hours. GMWR_1H consistently demonstrates the largest RMSE reduction for relative humidity. However, it should be noted that the direct assimilation of GMWR data caused a negative impact on relative humidity at a lead time of 12 hours. The degradation of wind fields (Fig. 9) and the model's inherent nonlinearity may be responsible."

8. Line 338-339: "likely due to a gradual increase in model error". This is not accurate, as this happens to both CNTL and MWR.

Response:

As integration time increases, systematic model errors accumulate. The atmosphere is highly nonlinear, and initial and model errors may not combine linearly, potentially filtering out improvements in the initial state. In addition, the assimilation algorithm is imperfect and does not improve all variables across all regions (e.g., the wind field shown in Fig. 9), while multivariate interactions may further diminish the overall effect. The explanation previously provided in the manuscript may not be sufficiently accurate. As this aspect is not the focus of this study, this sentence has been removed to avoid potential misinterpretation.

9. Line 342: "can further improve forecasts" -> "can further improve the short-term forecasts".

Response:

Corrected

10. Line 362-366: "This may be attributed to … the first 12 hours".    This logic in this part does not read good, and it looks like they can be removed.

Response:

We appreciate the reviewer's comment and agree that the original sentence lacked clarity and was prone to misinterpretation. The primary point we intended to convey is that the relatively limited improvements observed in the figure could be related to the long forecast lead times (12 and 24 hours), during which model errors tend to accumulate and diminish the benefits of improved initial conditions provided by GMWR assimilation. The sentences have been revised as follows:

"The limited improvement shown in this figure could be related to the relatively long forecast lead times (12 and 24 hours), during which model errors tend to accumulate and weaken the benefits of improved initial conditions from GMWR assimilation. Verification against surface station observations indicates that the improvements were primarily confined to the first few hours, particularly for temperature and humidity. After 12 hours, the impact declined noticeably, with some cases even exhibiting negative effects (Fig. 11)."

11. There are quite a few locations where articles "the/a" are missing (some of them are mentioned below). Consider a thorough proofreading of the manuscript.

Response:

We have thoroughly proofread the manuscript and corrected all identified instances where articles ("the/a") were missing. We also performed an additional grammar check to ensure clarity and correctness throughout the text.

Edits:

12. Line 50: The first appearance of "RTTOV", please define this acronym.

Response:

The full name of RTTOV has been defined at its first occurrence in the revised manuscript.

13. Fig. 1: The legend says "SOUND", suggest changing to "SONDE" or "SOUNDING"

Response:

  Corrected. The legend label has been changed from "SOUND" to "SONDE".

14. Line 73: remove "Similarly, "

Response:

  Corrected.

15. Line 105: change to "into the radiance space"

Response:

  Corrected.

16. Fig 2: Suggest removing the thumbnail figures in Fig. 2e and 2f; put the green lines into Fig. 2c and 2d

Response:

  Corrected.

17. Fig. 116-157: Shortening this sentence to "an MWR direct assimilation module was developed within WRFDA"

Response:

  Corrected.

18. Line232: change to "the above bias correction model,"

Response:

  Corrected.

19. Line 354: "against station observations" -> "against surface station observations". Change other occurrences of "station observations" throughout the manuscript.

Response:

  Revised as suggested. All instances of "station observations" have been changed to "surface station observations".

20. Line 416: "typically exceeds 700 hPa" -> "is typically lower than 700 hPa"
Response:

  Corrected.

21. Line 429: "This type covariances" -> "This type of covariances"
Response:

  Corrected.

---

## Referee Report (RR1)

Review of

Direct assimilation of ground-based microwave radiometer observations with machine learning bias correction based on developments of RTTOV-gb v1.0 and WRFDA v4.5

Overall, this paper is clearly written, including detailed descriptions of the experimental design. The analysis and presentation of the results are also generally clear. However, I have some concerns regarding the bias correction method and the way the experiments are compared. I would suggest that the authors address these points before the paper can be considered for publication.

Major comments:

1. The paper includes a focus on bias correction, but its impact on the analysis and forecast fields is not discussed in depth. While the improvements in GMWR diagnostics in observation space (e.g., Figs. 6–8) are somewhat expected, the more crucial aspect is how the bias correction affects the model space. It would strengthen the paper to include a comparison between assimilation experiments with and without bias correction for GMWR.

2. The bias correction approach based on offline O–B statistics essentially assumes that all biases originate from the observations. However, this assumption may not always hold, and such correction could potentially mask model bias (e.g., Auligné et al., 2007; Eyre, 2016), especially when more complex predictors or bias-prediction schemes are used to make the correction more expressive. Therefore, such offline bias correction that simply brings the O–B mean close to zero does not necessarily indicate a successful correction. It may be a sign of success, but could also be a result of compensating for model bias, rather than removing observation bias. Although in practice it remains difficult to fully separate the sources of O–B bias, I believe it is important for the paper to acknowledge this fundamental limitation of the current offline bias correction method based on O-B statistics.

Auligné, T., McNally, A. P., & Dee, D. P. (2007). Adaptive bias correction for satellite data in a numerical weather prediction system. *Quarterly Journal of the Royal Meteorological Society: A journal of the atmospheric sciences, applied meteorology and physical oceanography*, *133*(624), 631-642.

Eyre, J. R. (2016). Observation bias correction schemes in data assimilation systems: A theoretical study of some of their properties. *Quarterly Journal of the Royal Meteorological Society*, *142*(699), 2284-2291.

3. Regarding the wind analysis and forecast, the results appear somewhat inconsistent depending on the diagnostic used. For example, Fig. 9 shows a degradation, while Figs. 10–11 indicate marginal improvements. However, the paper does not seem to acknowledge or discuss these discrepancies across different diagnostics. A brief discussion of these differences would help clarify the interpretation of the results.

Minor Comments:

1. Figure 3(b)(d): It would be useful to show the same scatter plot after bias correction to examine whether the two distinct clusters merge.

2. L233-235: Existing approaches can also address nonlinear relationship between the physical variables, e.g., skin temperature (TS), and the bias since the selection of predictors can be completely general. E.g., consider predictors $p_1 = TS$, $p_2 = (TS)^2$, $p_3 = (TS)^3$, $p_4 = (TS)^4$...

3. L326-327: Since Figure 8 does not show the CTRL results, it is difficult to determine whether the assimilation of GMWR really improves the fit (even though such improvement is expected)

4. L480-481: This may not be true, as discussed in the major comment (2).

5. L498-499: A larger STD in the K-band compared to the V-band does not necessarily imply that the model's humidity accuracy is worse than its temperature accuracy. First, it is inherently difficult to directly compare the accuracy of humidity and temperature fields. Second, the brightness temperature STD also depends on its sensitivity to temperature and humidity. For example (using hypothetical numbers), a 1K change in K-band may correspond to 1% change in humidity, but 1K change in V-band may correspond to a much larger 10% change in temperature.

6. Overall, the paper includes a large amount of numerical detail (e.g., bias reductions by a few degrees or a certain percentage). If some of these values are already shown in the figures, I believe it is not necessary to restate all of them in the text. Instead, the paper could focus on highlighting the meaning and implications of these numbers. This would help make the manuscript more concise and easier to follow.

---

## Author Response (AR2)

Dear Reviewers:

We sincerely appreciate your valuable comments and suggestions, which have significantly improved the quality of our study. We have revised the manuscript accordingly and responded to all comments. Below are our point-by-point responses. The reviewers' comments are presented in black, our responses in blue, and the proposed changes to the manuscript in red.

**Reply to Reviewer #1**:

Overall, this paper is clearly written, including detailed descriptions of the experimental design. The analysis and presentation of the results are also generally clear. However, I have some concerns regarding the bias correction method and the way the experiments are compared. I would suggest that the authors address these points before the paper can be considered for publication.

Major comments:

1. The paper includes a focus on bias correction, but its impact on the analysis and forecast fields is not discussed in depth. While the improvements in GMWR diagnostics in observation space (e.g., Figs. 6–8) are somewhat expected, the more crucial aspect is how the bias correction affects the model space. It would strengthen the paper to include a comparison between assimilation experiments with and without bias correction for GMWR.

Response:

Thank you for your valuable suggestion. Based on the 1-hour assimilation interval experiment (GMWR_1H) already presented in the manuscript, an additional experiment (GMWR_1H_noBC) without bias correction (BC) was conducted. As shown in Figure R1, the initial conditions were verified against radiosonde observations. The results indicate that the experiment with BC reduced the RMSE of temperature and water vapor in the lower atmosphere. In contrast, the experiment without BC yielded only minor improvements in temperature RMSE and even increased the RMSE for water vapor. Specifically, for temperature below 1 km, the experiment with BC reduced the RMSE by

6.32%, whereas the experiment without BC achieved only a 0.49% reduction. For water vapor below 5 km, the experiment with BC reduced the RMSE by 1.98%, while the experiment without BC resulted in an 8.47% increase.

Similarly, as shown in Figure R2, the forecast fields were verified against surface station observations. The results show that the experiment with BC reduced the RMSE of 2m temperature and relative humidity at a lead time within 12 hours. In contrast, the experiment without BC degraded the forecast accuracy of 2 m temperature and relative humidity. Specifically, the time-averaged temperature RMSE decreased by 0.019 K with BC, whereas it increased by 0.001 K without BC. For relative humidity, the time-averaged RMSE decreased by 0.102 with BC, while it increased by 0.079 without BC.

These findings suggest that omitting BC weakens the effectiveness of assimilation and may even introduce negative impacts on both the initial and forecast fields. The above results have been covered in the discussion section, as shown below.

"The machine learning-based BC scheme effectively mitigated the bimodal distribution and systematic errors in the O−B statistics. To assess its impact on the initial and forecast fields, a parallel experiment without bias correction was conducted, based on the 1-hour assimilation interval experiment (GMWR_1H). For the initial fields, as verified against radiosonde observations, the experiment without BC yielded only minor improvements in temperature and even degraded the water vapor field. As for the forecast fields, verification against surface station observations showed that the absence of BC led to a noticeable degradation in the forecast accuracy of 2 m temperature and relative humidity. These findings indicate that the machine learning-based BC scheme had a beneficial impact on both the initial conditions and the subsequent forecasts."

[Figure]

Figure R1. Verification of the initial conditions against radiosonde observations, based on the ten-day assimilation experiment conducted from 13 to 22 October 2023. Differences in root mean square error (RMSE) for (a) temperature and (b) water vapor mixing ratio between experiments.

[Figure]

Figure R2. Verification of the forecast against surface station observations, based on the ten-day assimilation experiment conducted from 13 to 22 October 2023. Differences in root mean square error (RMSE) for (a) temperature and (b) relative humidity between experiments.

2. The bias correction approach based on offline O–B statistics essentially assumes that all biases originate from the observations. However, this

assumption may not always hold, and such correction could potentially mask model bias (e.g., Auligné et al., 2007; Eyre, 2016), especially when more complex predictors or bias-prediction schemes are used to make the correction more expressive. Therefore, such offline bias correction that simply brings the O–B mean close to zero does not necessarily indicate a successful correction. It may be a sign of success, but could also be a result of compensating for model bias, rather than removing observation bias. Although in practice it remains difficult to fully separate the sources of O–B bias, I believe it is important for the paper to acknowledge this fundamental limitation of the current offline bias correction method based on O-B statistics.

Auligné, T., McNally, A. P., & Dee, D. P. (2007). Adaptive bias correction for satellite data in a numerical weather prediction system. Quarterly Journal of the Royal Meteorological Society: A journal of the atmospheric sciences, applied meteorology and physical oceanography, 133(624), 631-642.

Eyre, J. R. (2016). Observation bias correction schemes in data assimilation systems: A theoretical study of some of their properties. Quarterly Journal of the Royal Meteorological Society, 142(699), 2284-2291.

Response:

We sincerely appreciate your valuable suggestion. Following Zhang et al. (2023, 2024), a nonlinear bias correction scheme based on O–B statistics was applied in this study. The assimilation experiment incorporating this bias correction demonstrated improvements in both the initial and forecast fields. However, as you rightly pointed out, the background field inherently contains biases, and the use of an offline correction method may potentially mask these model bias. In response to your suggestion, we conducted comparative experiments with and without this bias correction scheme. Although the results show that the bias correction had a positive impact on both the initial conditions and forecasts, it is admitted that this approach has some limitations in differentiating between model- and observation-related components of systematic biases.

This limitation has now been clearly stated in the discussion follows:

"Nevertheless, despite its demonstrated effectiveness, the scheme is subject

to several limitations. Relying on offline O–B statistics, it implicitly assumes that all biases originate from the observations—an assumption that may not always hold and may, in some instances, mask model biases (Auligné et al., 2007; Eyre, 2016). Moreover, the offline scheme lacks anchoring observations, rendering the analysis fields more susceptible to model bias. Future efforts should consider bias correction strategies based on unbiased reference observations or adopt a constrained correction scheme, such as the constrained adaptive bias correction (Han and Bormann, 2016)."

Zhang, X., D. Xu, X. Li, and F. Shen, 2023: Nonlinear Bias Correction of the FY-4A AGRI Infrared Radiance Data Based on the Random Forest. *Remote Sensing*, **15**, 1809, https://doi.org/10.3390/rs15071809.

Zhang, X., D. Xu, F. Shen, and J. Min, 2024: Impacts of Offline Nonlinear Bias Correction Schemes Using the Machine Learning Technology on the All-Sky Assimilation of Cloud-Affected Infrared Radiances. *Journal of Advances in Modeling Earth Systems*, **16**, e2024MS004281, https://doi.org/10.1029/2024MS004281.

3. Regarding the wind analysis and forecast, the results appear somewhat inconsistent depending on the diagnostic used. For example, Fig. 9 shows a degradation, while Figs. 10–11 indicate marginal improvements. However, the paper does not seem to acknowledge or discuss these discrepancies across different diagnostics. A brief discussion of these differences would help clarify the interpretation of the results.

Response:

Thanks for your comments. The discussion of these differences has been added to the discussion as follows:

"Although the assimilation of GMWR radiance yields slight improvements in the forecast wind fields (Fig. 11), it exerts an overall negative impact on the wind fields in the initial conditions (Fig. 9). This discrepancy stems from the use of distinct observational datasets: the initial conditions are verified against radiosonde observations, while the forecasts are evaluated using surface station data. It should be noted that assimilating GMWR improves the wind field below 500 m AGL in the initial conditions. This improvement is consistent with the

verification of the forecast, which demonstrate enhancements in the 2-meter wind fields. Regarding the degradation for wind field above 500m AGL, the background error covariance may contribute to this negative impact. On the one hand, it determines the response of the wind fields to temperature and humidity adjustments made by RTTOV-gb. On the other hand, since RTTOV-gb's adjustments are primarily concentrated in the ABL, the response above the ABL may be propagated through the background error covariance."

Minor Comments:

1. Figure 3(b)(d): It would be useful to show the same scatter plot after bias correction to examine whether the two distinct clusters merge.

Response:

Thank you for your suggestion. The two distinct clusters can result in a bimodal distribution in the O−B probability density function (PDF). For example, a cluster shifted to the right of the diagonal in the scatter plot leads to a peak on the positive x-axis of the O−B PDF. In the "Bias Correction" section, the bias correction model was tested. As shown in Figure 6b, d, the O−B PDF transitions from a bimodal to a unimodal distribution, indicating that the two clusters are effectively merged. Figure 6 is used to illustrate this issue rather than adding an additional scatter plot, as the manuscript already includes 17 figures. The corresponding revisions we have implemented are as follows:

"The bimodal feature in the O–B PDF corresponds to the two distinct clusters observed in the scatter plots shown in Fig. 3. Specifically, when one cluster is concentrated along the diagonal and the other shifts to the right, a peak forms on the positive x-axis of the O−B PDF, resulting in a bimodal distribution. From the O–B PDF (Fig. 6), both instruments exhibit a positive bias with a unimodal distribution in the K-band. In contrast, the V-band displays a bimodal distribution: the second peak appears on the right for HATPRO and on the left for MP3000A. These results are consistent with the scatter plots shown in Fig. 3. After BC, the O–B PDF changes from a bimodal to a unimodal distribution,

indicating that the two clusters have been effectively merged.

[Figure]

Figure 6: Probability density functions (PDFs) of the O−B distributions, based on a test set randomly selected from 30% of the three-month sample dataset collected from August to October 2023. The top and bottom rows correspond to the HATPRO and MP3000A sensors, respectively, while the left and right columns represent the K-band and V-band. Each panel displays the number of samples (num), the mean (bias), standard deviation (STD), skewness, and kurtosis of the distributions."

2. L233-235: Existing approaches can also address nonlinear relationship between the physical variables, e.g., skin temperature (TS), and the bias since the selection of predictors can be completely general. E.g., consider predictors $p1 = TS, p2 = (TS)^2, p3 = (TS)^3, p4 = (TS)^4 \dots$

Response:

Thank you for your comment. Most offline bias correction methods assess

the relationship between biases and predictors using multivariate linear regression. Although incorporating nonlinear operators into physical variables can enhance their ability to handle nonlinear relationships, machine learning approaches provide a more efficient means of capturing such complexities. The corresponding revisions we have implemented are as follows:

"O–B bias is commonly represented using multiple linear regression with several predictors. Compared to linear estimates, nonlinear approaches show improved performance in reducing systematic biases (Zhang et al., 2023, 2024). Following these works, this study employed a machine learning–based bias correction scheme using the Random Forest (RF) technique (Breiman, 2001)."

3. L326-327: Since Figure 8 does not show the CTRL results, it is difficult to determine whether the assimilation of GMWR really improves the fit (even though such improvement is expected)

Response:

Following your suggestion, we have included the results of the CNTL experiment. The corresponding revisions in the manuscript are as follows:

"O–A statistics were also computed based on the initial fields from the CNTL experiment. The O–A bias in CNTL is slightly larger than that in the GMWR assimilation experiments, with a more noticeable difference in the V band. Regarding the STD, the CNTL experiment shows higher values across all channels, with the largest difference approaching 1 K. These results suggest that assimilating GMWR data improves the consistency between the initial fields and the observed brightness temperatures.

[Figure]

Figure 6: Verification of the initial conditions against GMWR observations, based on the ten-day assimilation experiment conducted from 13 to 22 October 2023. (a) Bias and (b) standard deviation (STD) of the observation minus background (O−B) and observation minus analysis (O−A) for the GMWR assimilation in the target region of Southwest China (blue box in Fig. 1)."

4. L480-481: This may not be true, as discussed in the major comment (2).

Response:

The corresponding modifications are as below.

"After applying this BC model, both the bias and STD of the O−B were substantially reduced. Specifically, the bias and STD decreased by 0.83 K (97.1 %) and 1.63 K (64.6 %), respectively. For some channels, the original O−B distribution exhibited a bimodal pattern, which was transformed into a unimodal distribution after BC. The corrected O−B distributions exhibited Gaussian characteristics centered around zero."

5. L498-499: A larger STD in the K-band compared to the V-band does not necessarily imply that the model's humidity accuracy is worse than its

temperature accuracy. First, it is inherently difficult to directly compare the accuracy of humidity and temperature fields. Second, the brightness temperature STD also depends on its sensitivity to temperature and humidity. For example (using hypothetical numbers), a 1K change in K-band may correspond to 1% change in humidity, but 1K change in V-band may correspond to a much larger 10% change in temperature.

Response:

Thank you for your comment The corresponding sentence has been removed.

6. Overall, the paper includes a large amount of numerical detail (e.g., bias reductions by a few degrees or a certain percentage). If some of these values are already shown in the figures, I believe it is not necessary to restate all of them in the text. Instead, the paper could focus on highlighting the meaning and implications of these numbers. This would help make the manuscript more concise and easier to follow.

Response:

Thank you for your valuable comment. We have carefully revised the manuscript by removing redundant numerical details. These revisions enhance the manuscript's conciseness and clarity, making it easier to follow and more focused on the key findings.

**Reply to Reviewer #2**:

This is a high-quality and well-structured manuscript that makes a valuable contribution to the field of data assimilation. The authors present an effective and practical approach for the direct assimilation of ground-based microwave radiometer observations, supported by a carefully designed bias correction scheme. The results are clearly presented and demonstrate consistent improvements in low-level thermodynamic fields and short-term forecasts, particularly when a higher assimilation frequency is used. This study complements recent work in the field and offers useful insights for future operational applications. Nevertheless, several specific issues should be addressed before the manuscript can be considered for publication in GMD.

General comments,

1. Please discuss what is the physical significance of systematic bias? Does and offline or online BC conducted? If it is an offline bias correction, without the presence of an anchoring observation, how do you expect to differentiate the model vs observation component of the systematic biases? More physical interpretation would be helpful.

Response:

Thank you for your comments. Following Zhang et al. (2023, 2024), a nonlinear BC scheme was employed in this study. This approach is based on O–B statistics and assumes that all biases originate from the observations. Due to this limitation, it may mask model biases and fails to differentiate the model vs observation component of the systematic biases.

Nevertheless, additional experiments without BC were conducted to further investigate the impact of this BC method (In the response to Reviewer #1). Based on the 1-hour assimilation interval experiment (GMWR_1H) already presented in the manuscript, an additional experiment (GMWR_1H_noBC) without bias correction (BC) was conducted. As shown in Figure R1, the initial conditions were verified against radiosonde observations. The results indicate that the experiment with BC reduced the RMSE of temperature and water vapor in the lower atmosphere. In contrast, the experiment without BC yielded only minor

improvements in temperature RMSE and even increased the RMSE for water vapor. Specifically, for temperature below 1 km, the experiment with BC reduced the RMSE by 6.32%, whereas the experiment without BC achieved only a 0.49% reduction. For water vapor below 5 km, the experiment with BC reduced the RMSE by 1.98%, while the experiment without BC resulted in an 8.47% increase. Similarly, as shown in Figure R2, the forecast fields were verified against surface station observations. The results show that the experiment with BC reduced the RMSE of 2m temperature and relative humidity at a lead time within 12 hours. In contrast, the experiment without BC degraded the forecast accuracy of 2 m temperature and relative humidity. Specifically, the time-averaged temperature RMSE decreased by 0.019 K with BC, whereas it increased by 0.001 K without BC. For relative humidity, the time-averaged RMSE decreased by 0.102 with BC, while it increased by 0.079 without BC.

The results demonstrate that, in the absence of bias correction, GMWR assimilation becomes less effective and may even negatively impact both the initial and forecast fields. As noted in major comment (2) by Reviewer #1, it remains difficult in practice to separate the sources of O–B bias. Although this limitation exists in the current BC method based on O–B statistics, it still exerts a beneficial impact on GMWR assimilation, improving both the initial conditions and subsequent forecasts. As an initial study primarily focused on the direct variational assimilation of GMWR radiance, it is admitted that this study has certain limitations. Nevertheless, we have acknowledged the limitations the BC approach employed in this study and have proposed directions for future improvement. The corresponding statement has been included in the discussion section, as shown below.

"The machine learning-based BC scheme effectively mitigated the bimodal distribution and systematic errors in the O−B statistics. To assess its impact on the initial and forecast fields, a parallel experiment without bias correction was conducted, based on the 1-hour assimilation interval experiment (GMWR_1H). For the initial fields, as verified against radiosonde observations, the experiment without BC yielded only minor improvements in temperature and even degraded

the water vapor field. As for the forecast fields, verification against surface station observations showed that the absence of BC led to a noticeable degradation in the forecast accuracy of 2 m temperature and relative humidity. These findings indicate that the machine learning-based BC scheme had a beneficial impact on both the initial conditions and the subsequent forecasts. Nevertheless, despite its demonstrated effectiveness, the scheme is subject to several limitations. Relying on offline O–B statistics, it implicitly assumes that all biases originate from the observations—an assumption that may not always hold and may, in some instances, mask model biases (Auligné et al., 2007; Eyre, 2016). Moreover, the offline scheme lacks anchoring observations, rendering the analysis fields more susceptible to model bias. Future efforts should consider bias correction strategies based on unbiased reference observations or adopt a constrained correction scheme, such as the constrained adaptive bias correction (Han and Bormann, 2016)."

[Figure]

Figure R1. Verification of the initial conditions against radiosonde observations, based on the ten-day assimilation experiment conducted from 13 to 22 October 2023. Differences in root mean square error (RMSE) for (a) temperature and (b) water vapor mixing ratio between experiments.

[Figure]

Figure R2. Verification of the forecast against surface station observations, based on the ten-day assimilation experiment conducted from 13 to 22 October 2023. Differences in root mean square error (RMSE) for (a) temperature and (b) relative humidity between experiments.

Zhang, X., D. Xu, X. Li, and F. Shen, 2023: Nonlinear Bias Correction of the FY-4A AGRI Infrared Radiance Data Based on the Random Forest. *Remote Sensing*, **15**, 1809, https://doi.org/10.3390/rs15071809.

Zhang, X., D. Xu, F. Shen, and J. Min, 2024: Impacts of Offline Nonlinear Bias Correction Schemes Using the Machine Learning Technology on the All-Sky Assimilation of Cloud-Affected Infrared Radiances. *Journal of Advances in Modeling Earth Systems*, **16**, e2024MS004281, https://doi.org/10.1029/2024MS004281.

2. Section 3.1 Bias correction scheme BASE on OMB, is not designed to remove the model biases because if model background biases generate the OmB systematic biases, you not only need to correct the biases in OmBs but also in the background. Zhu et al. (2014) stated that the error due to the background atmospheric profiles should not be removed by the bias correction. Please clarify.

Response:

Thank you for your professional comments. The bias correction scheme based on O–B statistics has some limitations: it cannot differentiate between model- and observation-related components of systematic biases. As a result, the

approach may mask model biases and produce biased analyses. On the one hand, model biases should be addressed separately prior to data assimilation. On the other hand, it may be possible to consider bias correction scheme based on unbiased reference observations or adopt a constrained correction scheme, such as the constrained adaptive bias correction (Han and Bormann, 2016). However, despite these limitations, the current BC method based on O–B statistics still exerts a beneficial impact on GMWR assimilation, improving both the initial conditions and subsequent forecasts. Separating model and observation biases is a valuable research topic but beyond the scope of this initial study. In future work, we plan to explore a more effective bias correction strategy to further improve the GMWR assimilation. The relevant revision has been added to the discussion section, consistent with the response to general comment 1 above.

Han W. and Bormann N.: Constrained adaptive bias correction for satellite radiance assimilation in the ECMWF 4D-Var system, *ECMWF Technical Memoranda*, 783, https://doi.org/10.21957/rex0omex, 2016

Specific comments:

1. The description of the single-observation assimilation experiment could be more detailed, including information such as the instrument used, and which channels were assimilated.

Response:

The corresponding statements have been updated as below.

"A single-observation assimilation experiment was conducted to test the GMWR direct assimilation module. In this experiment, a pseudo radiance observation from the HATPRO sensor was assimilated in channel 6 (K band), with an assigned observation error of 1 K and an innovation of 2 K."

2. The hyperparameters of the machine learning bias correction model are important. As the figure related to hyperparameter tuning is currently omitted from the manuscript, consider including it—either in the main text or the

appendix. Alternatively, providing the tuning range and final configuration would be helpful as a reference for future studies.

Response:

The corresponding statements have been updated as below.

"During hyperparameter tuning, *n_estimators* was varied between 10 and 150, *max_depth* was adjusted from 5 to 30, *min_samples_leaf* was tested with values between 1 and 3, and *min_samples_split* was tuned in the range of 2 to 6.

The number and depth of trees are critical hyperparameters that must be predetermined. Training time increases approximately linearly with the number of trees, while performance exhibits a logarithmic-like saturation trend. In term of tree depth, both training time and performance increase approximately logarithmically with depth. Thus, selecting a modest number (*n_estimators*) and depth (*max_depth*) of trees, such as 50 and 15, can balance efficiency and accuracy."

3. Figure 1: The legend appears to be incorrect. It should be "GMWR" instead of "MWR"

Response:

Corrected.

4. Figure A2: The instrument label is incorrect. Both Figure A1 and A2 show the sensor as "HATPRO".

Response:

The label in Figure A2 should be "MP3000A", and now is updated.

5. Line 45: Change "observation-operator" to "observation operators."

Response:

Corrected.

6. Line 48: Change "It is noted that studies began to develop fast RTMs suitable for GMWR" to "Recent studies have developed fast RTMs suitable for GMWR."

Response:

The corresponding sentences has been updated.

7. Line 98: Change "dictates" to "dictate."

Response:

Corrected.

8. Lines 159–166: Pay attention to capitalization. In "a machine learning bias correction scheme" consider whether "a" should be capitalized to be consistent with formatting.

Response:

Corrected.

9. Line 187: Add a period before "Therefore."

Response:

Added.

10. Line 191: Change "only calculate O−B" to "only calculating O−B."

Response:

Corrected.

11. Lines 238–240: There should be a space between the numeric value and "hPa". Please check the manuscript for similar issues with other units

Response:

Thanks for your comments. We have thoroughly reviewed the entire manuscript.

12. Line 287: Change "scattering patterns" to "scatter patterns."

Response:

Thanks for your comments. The relevant sentence has already been revised in response to other comments, and this issue no longer exists

13. Line 471: An article (e.g., the or a) should be added before "GMWR assimilation module".

Response:

Added.

14. Line 474: The tense of "developed" appears to be inconsistent with "evaluate" and should be adjusted for parallel structure.

Response:

The corresponding statements have been updated as below.

"Based on this module, a three-month sample of O−B statistics was calculated to evaluate the bias and develop a BC model."

15. Line 475: Change "direct assimilation GMWR" to "direct assimilation of GMWR."

Response:

The corresponding statements have been updated as below.

"Furthermore, 10-day assimilation experiments (Table 2) were conducted using this GMWR assimilation module and BC model to investigate the impact of direct GMWR assimilation and the effects of assimilation frequency."

16. Lines 490–491: Change "has potential to" to "has the potential to".

Response:

Corrected.

---

## Author Response (AR3)

Dear Reviewers:

We sincerely appreciate your valuable comments and suggestions, which have significantly improved the quality of our study. We have revised the manuscript accordingly and responded to all comments. Below are our point-by-point responses. The reviewers' comments are presented in black, our responses in blue, and the proposed changes to the manuscript in red.

**Reply to Reviewer #1**:

Thanks for addressing previous comments. The latest version has been greatly improved.

I have the following minor comments:

1. Line 145: "and is generally consistent with Chinese administrative divisions": It looks like this sentence is not needed and can be removed.

Response:

Thank you for your comment. This sentence has been removed.

2. Line 159: "34 (0.91%), 70 (1.42%), and 76 (0.72%) observations were rejected" -> why does a 6h window reject fewer observations with a larger percentage while a 1h window rejects more observations but with a smaller percentage?

Response:

Thank you for your comment. This study uses a 12-hour 3DVAR cycling assimilation framework, where the different experiments correspond to cycling intervals of 6 hours, 3 hours, and 1 hour for assimilating GMWR observations. In each assimilation cycle, only the MWR observations closest to the analysis time within ±10 minutes are selected. Therefore, the 1-hour cycling experiment includes a much larger number of available GMWR observations overall. As a result, although the total number of rejected observations increases, their proportion relative to all assimilated observations becomes smaller. For greater clarity, the corresponding revisions are as follows:

"Each experiment started at 12:00 UTC daily, followed by a 12 h cycling

data assimilation period and a subsequent 24 h forecast.

Although the experiment with a 1 h assimilation interval rejects the largest number of observations, its rejection rate remains the lowest because it has the highest assimilation frequency and assimilates the largest volume of GMWR data."

3. Line 443-44: GMWR DA mainly improves forecasts 0~6h, so it is NOT that we cannot find improvements in the upper air, but we don't have radiosonde data to do the verification during corresponding period from 0~6h. One possible option to check the upper air forecast impact is to verify against aircraft data.

Response:

Thanks for your suggestion. We agree that GMWR DA mainly benefits 0–6 h forecasts, and the limited improvements indicated by the radiosonde-based evaluation should not be interpreted as "no improvement in the upper air." As an initial study, we primarily focus on implementing the direct assimilation of GMWR observations and demonstrating its potential; therefore, this work has some limitations. In the next step, within our planned satellite–ground combined assimilation framework, we will conduct a more comprehensive evaluation of the upper-air impacts using additional independent observations, including aircraft reports and radio occultation data. Nevertheless, the corresponding revisions in the radiosonde verification and discussion sections are as follows:

"Based on radiosonde-based verification at 12 and 24 h lead times, only limited improvements are evident after GMWR assimilation; however, this does not rule out larger impacts within 0–6 h, which cannot be robustly assessed here due to the limited temporal availability of radiosonde observations.

A more comprehensive evaluation of upper-air impacts could also be performed using additional independent observations, including aircraft reports and radio occultation data."

4. Line 450, Figure 12: Is this figure the average of forecasts at lead time of 12 and 24 hours? If yes, please be more descriptive in the figure caption.

Response:

Thank you for your question. Figure 12 shows the overall RMSE computed from all 12-h and 24-h forecast samples combined, rather than an average of individual forecast RMSEs. For greater clarity, the corresponding revisions are as follows:

"Figure 9. Verification of the initial conditions against radiosonde observations. RMSEs are computed from all samples over the ten-day assimilation experiment conducted from 13 to 22 October 2023.

Figure 12. Same as Fig. 9, but for forecasts at lead times of 12 and 24 h, with RMSEs computed from all forecast samples during the ten-day experiment."

5. Line 459: "relatively low FSS values observed around the 9 h forecast period": should it be "around the forecast hour 12" (based on Fig. 13)?

Response:

Corrected.

6. Fig. 13: Why is there a V shape in the FSS plots? Does it mean forecasts are dominated by another large scale feature?

Response:

Thank you for your comment. We re-checked the FSS calculation to clarify the origin of the V-shaped behavior. The V-shaped pattern is likely related to the diurnal variability of precipitation and a temporal phase mismatch between forecasts and observations (with the simulated precipitation occurring slightly earlier). Figure R1 shows the FSS together with the related diagnostic terms for the CNTL experiment. The FSS is defined as:

$$\text{FSS} = 1 - \frac{FBS}{PFO + POB}$$

In simple terms, PFO and POB indicate how prevalent/widespread the threshold-exceeding precipitation signal is in the forecast and observed fields,

respectively, while FBS measures how different the forecast and observed precipitation patterns are. Notably, POB reaches its minimum at the 12-h lead time, whereas PFO reaches its minimum at the 9-h lead time, indicating a lead-time (phase) mismatch in fractional coverage between forecasts and observations. Around the 12-h lead time, the overall event signal is relatively weak (small PFO+POB) but the forecast–observation difference is relatively large (large FBS), so the ratio FBS/(PFO+POB) becomes largest and the FSS reaches its minimum.

[Figure]

Figure R1. Time series of FSS, FBS, POB, and PFO from the CNTL experiment conducted from 13 to 22 October 2023. The scores were computed using 3-hour accumulated precipitation with a 3 mm threshold.

7. Line 469-470: The forecast verification against radiosonde does not agree with the conclusions here or that against surface observations.

Response:

Thank you for your comment. This sentence has been revised as follows:

"These findings are consistent with the above verification against surface station observations, suggesting that GMWR assimilation can improve forecasts

and that higher-frequency assimilation leads to further enhancements."

8. Line 503-509: This paragraph looks like some kind of duplicate with the paragraph in lines 482-487.

Response:

Thank you for pointing this out. Lines 503–509 partially overlapped with the results summarized in Lines 482–487. To avoid duplication, redundant content has been removed and the paragraph has been reorganized for clarity. The revised text is as follows:

"In the three-month O−B statistics, the STD in the K-band is larger than that in the V-band, consistent with Vural et al. (2024) and Cao et al. (2023). A notable positive O−B bias is observed at high-altitude stations over the Tibetan Plateau, which may be related to large-scale topographic effects. In this region, model simulations may contain errors, and RTTOV-gb coefficients may be inapplicable. The RTTOV-gb coefficients are based on global atmospheric profiles, which may differ significantly from the climatic conditions of plateau regions, potentially affecting simulation accuracy."

9. "5 Conclusion and Discussion": The latest version added lots of good discussions. But the logic in section 5 can be further improved to make it easier to follow by readers. One possible revision is to do the discussion first and then conclude with the 3 main findings and the future plan about "doing satellite-based radiance and ground-based ones together".

Response:

Thank you for your helpful suggestion. Following your suggestion, we split Section 5 into two subsections—Section 5.1 (Discussion) and Section 5.2 (Conclusions and future work)—to improve the logic and readability. The revised Section 5 is as follows:

"

[revised manuscript text omitted]

---

## Author Response (AR4)

Dear Reviewers:

We sincerely appreciate your valuable comments and suggestions, which have significantly improved the quality of our study. We have revised the manuscript accordingly and responded to all comments. Below are our point-by-point responses. The reviewers' comments are presented in black, our responses in blue, and the proposed changes to the manuscript in red.

**Reply to Reviewer #1**:

The latest version addressed most reviewers' comments. It is in a good shape for publication pending the following minor revisions:

1. Lines 511-512: "This discrepancy stems from... using surface station data". Suggest removing this sentence.

Response:

   Thank you for your comment. This sentence has been removed.

2. Lines 515-517: "On the one hand...on the other hand...": They talk about the same thing, i.e. background error covariance contribute to the negative impact which has been covered in Lines 514-515. Suggest removing them or revising them to read more smoothly.

Response:

   Thank you for the suggestion. These sentences (Lines 515–517) are intended to further explain the statement made in the previous sentence (Lines 514–515). We have revised the phrasing to make the connection smoother and clearer.

   "Regarding the degradation of wind fields above 500 m AGL, the background error covariance may contribute to this negative impact. Specifically, it propagates the RTTOV-gb increments concentrated in the ABL to higher levels and induces wind field adjustments in response to temperature and humidity updates."

3. Lines 563-564: 1. "Be leveraging their complementary...across multiple layers of the atmosphere": either remove this sentence or move it to before "a more comprehensive evaluation of ..."; 2. if moving this sentence, change "this approach has the potential to" to "we may" as well.

Response:

Thank you for the suggestion. We have chosen to remove this sentence as suggested.

---

## Author Response (AR5)

Dear Editor and the Production Team,

We sincerely appreciate the valuable comments from the reviewers and the professional support from the editor throughout the review process. Their contributions have been vital in refining our manuscript and have significantly improved the final quality of our study. We also thank the production team in advance for their assistance in bringing this work to publication.

While preparing the final production files, we would like to report a minor technical update to Figures 1 and 8. As the original editable source files were unfortunately lost, we carefully redrew these two figures to provide high-quality, production-ready versions. We confirm that these updates are purely graphical and do not affect any data, results, interpretations, or conclusions:

Figure 1: Updated the colorbar label from "m" to "Topography (m)" for clarity.

Figure 8: The legend was manually repositioned during redrawing, so its placement differs slightly from the previous version.

No other parts of the manuscript have been modified. Thank you for your assistance during the production process.

Best regards,
Qing Zheng (on behalf of all co-authors)